# High-resolution water level and storage variation datasets for 338 reservoirs in China during 2010–2021

Youjiang Shen[1], Dedi Liu[1], Liguang Jiang[2], Karina Nielsen[3], Jiabo Yin[1], Jun Liu[4], Peter Bauer-Gottwein[4]

[1]State Key Laboratory of Water Resources & Hydropower Engineering Science, Wuhan University, Wuhan, 430072, China
[2]School of Environmental Science and Engineering, Southern University of Science and Technology, Shenzhen, 518055, China
[3]DTU Space, National Space Institute, Technical University of Denmark, 2800, Kongens Lyngby, Denmark
[4]Department of Environmental Engineering, Technical University of Denmark, 2800, Kongens Lyngby, Denmark

*Correspondence to*: Dedi Liu (dediliu@whu.edu.cn)

**Abstract.** Reservoirs and dams are essential infrastructures in water management, thus information of their surface water area (SWA), water surface elevation (WSE), and reservoir water storage change (RWSC), is crucial for understanding their properties and interactions on hydrological and biogeochemical cycles. However, knowledge of these reservoir characteristics is scarce or inconsistent at national scale. Here, we introduce comprehensive reservoir datasets of 338 reservoirs in China, with a total of 470.6 km$^3$ storage capacity (50% Chinese reservoir storage capacity). Given the scarcity of publicly available gauged observations and operational applications of satellites for hydrological cycles, we utilize multiple satellite altimetry missions (SARAL/AltiKa, Sentinel-3 A and B, CroySat-2, Jason-3, and ICESat-2) and imagery data from Landsat and Sentinel-2 to produce a comprehensive reservoir dataset on the WSE, SWA, and RWSC during 2010–2021. Validation against gauged measurements of 93 reservoirs demonstrates the relatively high accuracy and reliability of our remotely-sensed datasets: (1) Across gauge comparisons of RWSC, the median statistics of Pearson correlation coefficient (CC), normalized root-mean-square error (NRMSE), and root-mean-square error (RMSE) are 0.89, 11%, and 0.021 km$^3$, with a total of 91% validated reservoirs (83 of 91) having good RMSE from 0.002 to 0.31 km$^3$ and NRMSE values smaller than 20%. (2) Comparisons of WSE retracked by six satellite altimeters and gauges show good agreement. Specifically, percentages of reservoirs having good and moderate RMSE values smaller than 1.0 m for CryoSat-2 (validated in 30 reservoirs), SARAL/AltiKa (9), Sentinel-3A (34), Sentinel-3B (25), Jason-3 (11), and ICESat-2 (26) are 77%, 75%, 79%, 87%, 81%, and 82% respectively. By taking advantages of six satellite altimeters, we are able to densify WSE observations across spatiotemporal scales. Statistically, around 96% validated reservoirs (71 of 74) have RMSE values below 1.0 m, while 57% reservoirs (42 of 74) have a good data quality with RMSE values below 0.6 m. Overall, our study fills such a data gap with regard to comprehensive reservoir information in China and provides strong support for many aspects such as hydrological processes, water resources, and other studies. The dataset is publicly available on Zenodo at https://doi.org/10.5281/zenodo.7251283 (Shen et al., 2021).

# 1 Introduction

Reservoirs and dams are essential infrastructures in water management that alters the natural river flows to provide services such as flood control, hydroelectricity generation and irrigation (Intralawan et al., 2018; Zhu et al., 2020). Largely mandated by the Flood Control Act of 1950, more than 98,000 reservoirs and dams have been constructed in China with a total water capacity of around 932 km$^3$ (Statistic Bulletin on China Water Activities, 2018). The boom of dam impoundment will continue for next decades in the background of climate warming and human activities (Lehner et al., 2011; Gutenson et al., 2020). Understanding the role of reservoirs and dams in the hydrological cycles has become increasingly important (Buccola et al., 2016; Marx et al., 2017; Chaudhari et al., 2018; Busker et al., 2019). Our review of the literature suggests that a combination of data and river models is the core to understand impacts of reservoirs on hydrological cycles. However, most studies on reservoirs have been significantly limited to data scarcity. Despite progress in process-based models with new reservoir schemes and hyper spatial resolution (Shin et al., 2019; Dang et al., 2020), most of them had approximated the reservoir releases just through storage-release equations and routed downstream with river routing mechanisms (e.g., Zhao et al., 2016; Zajac et al., 2017; Pokhrel et al., 2018; Yassin et al., 2019; Boulange et al., 2021). Acknowledging such approximations, we intend to contribute relevant studies by introducing the remotely-sensed reservoir datasets that can be applied as constraints to calibrate models or directly used for reservoir analysis. Our study plans to fill such a data gap, i.e., to develop the remotely-sensed reservoir datasets including surface water area (SWA), water surface elevation (WSE), and reservoir water storage change (RWSC) of 338 reservoirs in China.

Due to the absence of observational records describing the multitude of reservoir characteristics, remote sensing techniques have been developed to monitor reservoirs and have characterized reservoir across the globe (Gao et al., 2012; Duan and Bastiaanssen, 2013). Satellite missions have been used to offer reliable reservoir estimates such as SWA, WSE, and RWSC (Zhang et al., 2014; Wang et al., 2020; Zhang et al., 2020, Shen et al., 2022). WSE can be acquired by satellite laser or radar altimetry missions such as Sentinel-3A/B, CryoSat-2, Jason-1/2/3, and ICESat-1/2 (e.g., Wingham et al., 2006; Donlon et al., 2011; Zhang et al., 2011; Song et al., 2013; Jiang et al., 2020); SWA can be derived from SAR or optical images from e.g. MODIS, Landsat MSS/TM/OLI and Sentinel-1/2 (e.g., Goumehei et al., 2019; Weekley and Li, 2019); and RWSC can be calculated by two methods: one is using WSE and SWA from satellite altimeters and images, and the other one is using imagery-based SWA and digital elevation model (DEM). The core of these two methods is to construct the hypsometry relationships, i.e., Area-Elevation curves (A–E) from the overlapping records of WSE and SWA or DEM (Bonnema et al., 2016; Vu et al., 2022). There have been studies and online databases producing the remotely-sensed datasets for inland reservoirs/lakes at regional/global scales (Birkett et al., 2011; Crétaux et al., 2011; Gao et al., 2012; Zhang et al., 2014; Khandelwal et al., 2017; Getirana et al., 2018; Busker et al., 2019; Yao et al., 2019; Zhao and Gao, 2018; Li et al., 2020; Tortini et al., 2020). We have listed the studies and databases in Table 1 to summarize the progress of remotely-sensed data of reservoirs. Obviously, there is a data gap with regard to comprehensive reservoir information in China (Table A1). Records of a few reservoirs are available from these databases or previous studies (Table A1). Taking reservoir water level as an example,

approximately 30 Chinese reservoirs are available from three datasets (i.e., Hydroweb, G-REALM, and DAHITI). Therefore, studies dynamically incorporating various satellites into a comprehensive reservoir data set at national scale can fill the data gap.

With this motivation in mind, we further identified some limitations of the studies listed in Table 1. Most of them just focus on developing a single reservoir data set (WSE, SWA, RWSC, or A–E relationships) for a few reservoirs across the globe (Gao et al., 2012; Mu et al., 2020; Zhang et al., 2014). For example, Yao et al. (2019) constructs the long-term area time series for 428 reservoirs and lakes at bi-monthly scale by recovering inundation areas from contaminated Landsat-based images. The remotely-sensed products had often not been extensively validated by ground observed data, which are usually not publicly available, with the exception of a few studies with scare in-situ observations (Bonnema and Hossain, 2019). Khandelwal et al. (2017) mapped the global areal extent and temporal variations of reservoirs at 500 m spatial resolution, eight-day intervals from 2000 to 2015. Altimetric water level time series from 94 reservoirs were used to validate their area datasets due to the lack of in situ measurements. Tortini et al. (2020) provides a global data set of SWA, WSE, and storage change over 347 lakes/reservoirs, but results are validated at only one lake. Moreover, the remotely-sensed datasets (e.g., lake/reservoir storage variations by Busker et al., 2019 or RWSC by Avisse et al., 2017) are not publicly available. A geo-statistical approach has also been adopted to estimate RWSC with a surface water area during 1985–2005 (Fang et al., 2019), there are critical limitations shown as wide confidence intervals and high uncertainties due to its simplifications. There are several databases offering the time series of altimetry-derived WSE and/or imagery-based SWA estimates for big reservoirs across the globe. They are the Hydroweb (Crétaux et al., 2011), the G-REALM (Birkett et al., 2011), the DAHITI (Schwatke et al., 2015), the Bluedot observatory, RealSAT (Khandelwal et al., 2022) and others. These databases incorporated more altimetric information and provided datasets at higher temporal resolution. For example, Hydroweb firstly provided altimetry-derived water level time series on lakes and rivers from different satellite missions. Unlike Hydroweb, G-REALM focuses on some of the world's largest reservoirs/lakes. Within a rather unprecedented framework of online web application, Bluedot observatory allows for exploring and generating imagery-based SWA time series of reservoirs/lakes on-the-fly. As already mentioned, records of a few reservoirs in China are available. Whether reservoir WSE or SWA time series from these databases have a good agreement with one another and gauged measurements is not systematically evaluated, which can be shown in this study.

In light of the above, our objective is to fill research gap with regard to comprehensive reservoir information in China, thus supporting process-based models to better understand systematic reservoir effects. To densify reservoir observations, multiple satellite altimetry missions (i.e., Sentinel-3 A/B, SARAL/AltiKa, CroySat-2, Jason-3, and ICESat-2) and imagery data from Landsat and Sentinel-2 are utilized to develop high-resolution remotely-sensed reservoir datasets including SWA, WSE, and RWSC of 338 reservoirs in China during 2010–2021, with a total of 470.6 $km^3$ water capacity (50% reservoir water capacity in China). To validate the remotely-sensed results, the in-situ observations of 93 reservoirs are used for evaluations, thereby bringing the good level of confidence on the quality of datasets. Users are free to access datasets in an easily readable file format that allows researchers quickly handle our datasets at https://doi.org/10.5281/zenodo.7251283 (Shen et al., 2021).

Results of this study align with the efforts to understand role of reservoirs on hydrological cycles but significantly limited to data scarcity. Moreover, a growing interest in using remote sensing data in hydrological cycle is expected, thus knowing the accuracy of the remote sensing data is a prerequisite. Although previous studies assessed satellite altimeters in retrieving 100 reservoir water levels (Shu et al., 2021), knowledge is still limited as to evaluations of different altimeters for a large sample of reservoirs, which can be shown in this study. Overall, our unique contribution lies in the unique and novel remotely-sensed datasets to fill a data gap with regard to comprehensive reservoir information in China, and to benefit studies involving many fields such as hydrological processes, water resources, and other studies.

**Table 1. Summary of recent studies and databases producing the remotely-sensed data on WSE, SWA, RWSC, and hypsometric**
**curve of reservoirs.**

| Category | Product and reference | Source and remark |
|---|---|---|
| **WSE** | G-REALM, Birkett et al., 2011 | https://ipad.fas.usda.gov/cropexplorer/global_reservoir, reservoirs and lakes |
| | Hydroweb, Crétaux et al., 2011 | http://hydroweb.theia-land.fr/, for lakes and rivers |
| | Gao et al. 2012 | 34 global reservoirs, not publicly accessible |
| | DAHITI, Schwatke et al., 2015 | https://dahiti.dgfi.tum.de, rivers, and lakes/reservoirs |
| | AltEx, Markert et al., 2019 | https://altex.servirglobal.net, web application for exploring Jason and SARAL |
| | Tortini et al., 2020 | https://doi.org/10.5067/UCLRS-GREV2, 347 lakes and reservoirs |
| | Water level On VITO, CGLS | https://land.copernicus.eu/global/products/wl, lakes (~210) and rivers |
| **SWA** | Hydroweb, Crétaux et al., 2011 | http://hydroweb.theia-land.fr/, available for lakes |
| | Gao et al., 2012 | 34 global reservoirs, not publicly accessible |
| | Zhang et al., 2014 | 21 reservoirs, not publicly accessible |
| | DAHITI, Schwatke et al., 2015 | https://dahiti.dgfi.tum.de, lakes/reservoirs |
| | Khandelwal et al., 2017 | http://z.umn.edu/monitoringwaterRSE, 94 reservoirs |
| | GRASD, Zhao et al., 2018 | https://doi.org/10.18738/T8/DF80WG, 7,246 global reservoirs |
| | Busker et al., 2019 | 137 lakes and reservoirs, not publicly accessible |
| | Yao et al., 2019 | https://lakewatch.users.earthengine.app/view/glats, 205 reservoirs |
| | Liu et al., 2020 | 24 Chinese reservoirs, not publicly accessible |
| | Tortini et al., 2020 | https://doi.org/10.5067/UCLRS-AREV2, 347 lakes and reservoirs |
| | Donchyts et al., 2022 | https://doi.org/10.6084/m9.figshare.20359860, 71,208 lakes and reservoirs |
| | Khandelwal et al., 2022 | https://doi.org/10.5281/zenodo.4118463, 681,137 lakes and reservoirs |
| | Bluedot Observatory | https://blue-dot-observatory.com, available for lakes/reservoirs |
| **RWSC** | Gao et al., 2012 | 34 global reservoirs, not publicly accessible |
| | Zhang et al., 2014 | 21 reservoirs, not publicly accessible |
| | Busker et al., 2019 | 137 lakes and reservoirs, not publicly accessible |
| | DAHITI, Schwatke et al., 2020 | https://dahiti.dgfi.tum.de, 62 lakes/reservoirs |
| | Liu et al., 2020 | 24 Chinese reservoirs, not publicly accessible |
| | Tortini et al., 2020 | https://doi.org/10.5067/UCLRS-STOV2, 347 lakes and reservoirs |
| | Klein et al., 2021 | 1267 global reservoirs are analyzed, not publicly accessible |
| | Hou et al., 2022 | 6695 global reservoirs, not publicly accessible |
| | Vu et al., 2022 | https://doi.org/10.5281/zenodo.6299041, 10 reservoirs |
| **hypsometric curve** | Gao et al., 2012 | 34 reservoirs, not publicly accessible |
| | Zhang et al., 2014 | https://doi.org/10.1002/2014WR015829, 21 reservoirs |
| | Yigzaw et al., 2018 | http://wowuoh.wixsite.com/home/models-data, 6,800 reservoirs |
| | Vu et al., 2022 | https://doi.org/10.5281/zenodo.6299041, 10 reservoirs |
| **Our study** | | https://doi.org/10.5281/zenodo.7251283, 338 reservoirs with WSE, SWA, RWSC, and hypsometric curve during 2010-2021 in China |

\* Last access: 15 October 2022. Abbreviations are as follow: Global reservoir and dam database (GRanD), Database for hydrological time series of inland waters (DAHITI), Global reservoirs and lakes monitor (G-REALM), Global reservoir surface area dataset (GRSAD).

## 2 Data and Methods

China has an enormous network of reservoirs across different geographical landscapes. In this study, we selected all reservoirs for which geographical information is available from the GRanD database (http://globaldamwatch.org/grand/, Lehner et al., 2011). The GRanD provides an extensive number of attributes for reservoir shapefiles including geolocations of dams (i.e., latitude and longitude), extents and areas of reservoirs, dam heights, storage capacity and others. We found that there is a considerable variation in regulation capacity, water area, storage capacity, main function, and installed capacity to generate hydropower. Fig. 1 shows the spatial distribution of the final retained reservoirs, and the coverage of both altimetry passes and in-situ gauges as a reference for validation. Most reservoirs are densely grouped over eastern and middle China. These in-situ gauges provide a good testbed to evaluate performance of each altimetry over diverse reservoirs. We obtained daily water level and storage data spanning 2015–May 2021 for 93 reservoirs from the local watershed agency (http://xxfb.mwr.cn/index.html) and National Hydrological Information Centre for validation (http://113.57.190.228:8001/web/Report/BigMSKReport). All records follow a strict quality control, and the time series cover different periods. Data from May 2018 to June 2019 are missing for nearly all 93 reservoirs. 49 reservoirs cover the period of 2015–May 2021, while the remaining reservoirs cover the period of July 2019–May 2021. Each reservoir has a storage capacity more than 40 million $m^3$, with a total water capacity of 189.2 $km^3$. Detailed information about reservoirs with in situ data can be found in supplementary file (Fig. S1).

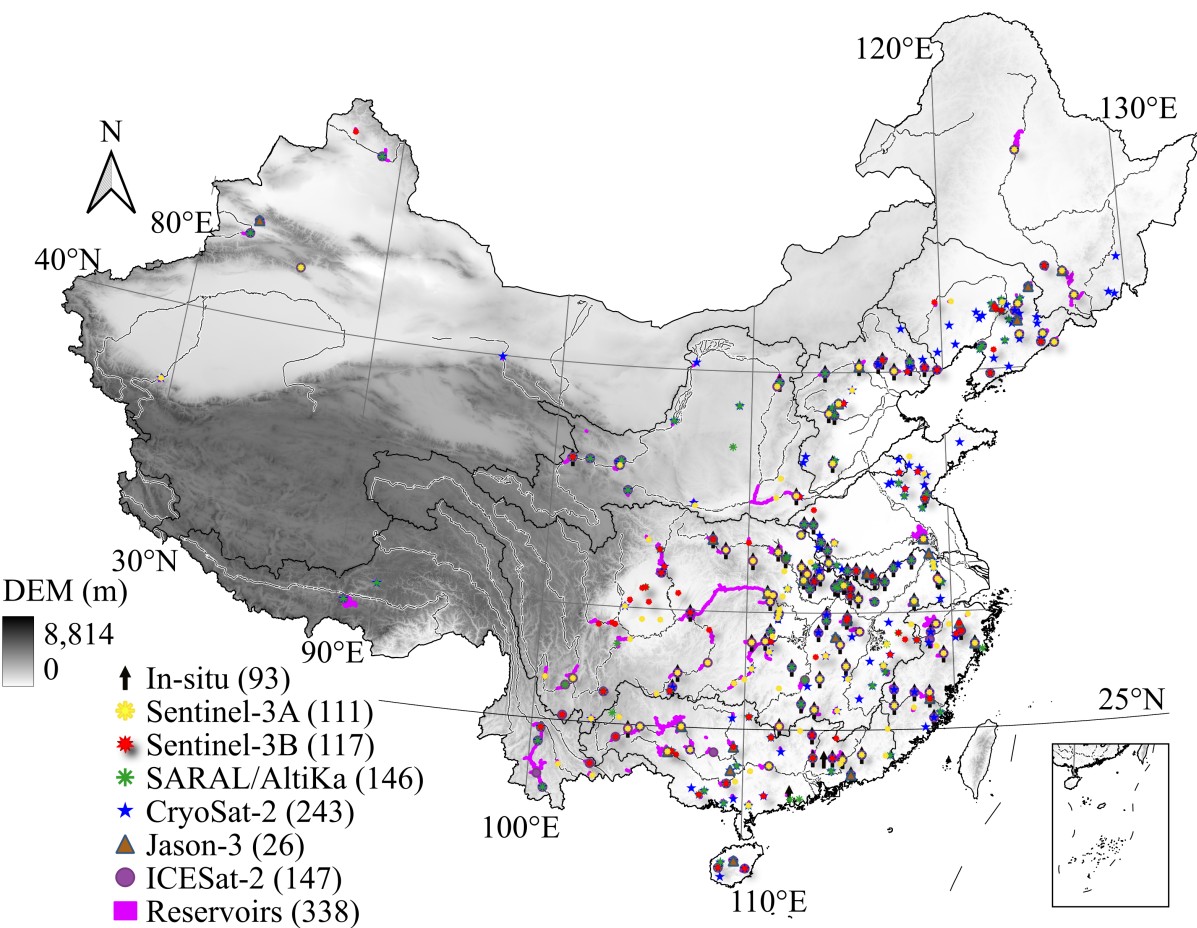

**Figure 1: Map of reservoirs covered by multisource satellite altimeters and stages. 338 reservoirs are finally retained in our datasets. For more details, please refer to Sect. 3.**

## 2.1 Satellite radar altimetry

We collect satellite altimetry-derived WSE measurements from CroySat-2, Sentinel-3 A/B, SARAL/AltiKa, Jason-3, and ICESat-2. For readers to get a broad understanding of these missions, the main features are summarized below while detailed information is available at the Official User Books (European Space Agency and Mullar Space Science Laboratory, 2012; Dinardo et al., 2018). CryoSat-2 (CS2) launched in April 2010 carries a Synthetic Aperture Interferometric Radar Altimeter. It operates in three modes, i.e., low resolution mode, synthetic aperture mode, and synthetic aperture interferometric mode. The baseline C level 1b dataset are from ESA (https://science-pds.cryosat.esa.int/), which provides 20 Hz measurements including waveforms, position, corrections, interferometric phase difference, etc. These waveforms were retracked with the Primary Peak Center Of Gravity (PPCOG), Narrow Primary Peak Threshold Retracker with a 50% (NPPTr[0.5]) and 80% (NPPTr[0.8]) threshold algorithms (Jain et al., 2015). SARAL/AltiKa launched in 2013 carries the first altimeter operating in Ka-band frequency, which enables a higher spatial resolution and leads to higher data availability (CNES, 2016). Note that

SARAL/AltiKa left its repetitive orbit with a repeat cycle of 35 days in July 2016 and switched to a drifting geodetic orbit with subcycles of 15-17 days with 1002 passes (Bonnefond et al., 2018). We downloaded their Geophysical Data Records (GDRs) from CNES (Centre National D'Etudes Spatiales) AVISO+ (Validation and Interpretation of Satellite Oceanographic data, ftp://avisoftp.cnes.fr/AVISO/pub/). The records provide 40 Hz ranges retracked using the Ice-1 and Ice-2 algorithms, and we also implemented the PPCOG, NPPTr[0.5] and NPPTr[0.8] algorithms to derive WSE. Sentinel-3 consists of the constellations of two satellites, i.e., Sentinel-3A (S3A) and Sentinel-3B (S3B), which were launched in February 2016 and April 2018, respectively. It is the first radar altimeter measuring in SAR mode at global scale with an open-loop tracking system (Biancamaria et al., 2018). It, therefore potentially facilitate more water level measurements at a higher precision an accuracy. We only used Ku-band SAR measurements with open-loop tracking mode and downloaded Level 2 "Enhanced measurements" datasets from https://scihub.copernicus.eu/dhus/. It contains 20 Hz measurements with waveforms, altitude, position, corrections, and ranges retracked by the Ocean algorithm. We also implemented the PPCOG, NPPTr[0.5] and NPPTr[0.8] algorithms, and the traditional Offset Center Of Gravity (OCOG) algorithms to derive water level. Jason-3 equipped with Poseidon-3B altimeter was launched in February 2016. It has an across-track resolution of 315 km at the equator, potentially offering altimetric data at a low spatial resolution but a high temporal resolution of 10 days. The GDRs of Jason-3 are downloaded from CNES AVISO+ (ftp://avisoftp.cnes.fr/AVISO/pub/), containing the ranges retracked by OCOG, Ocean, and Adapt algorithms. ICESat-2 launched in 2018 is carrying the Advanced Topographic Laser Altimeter System (ATLAS) and can provide detailed measurements of inland waters at an approximate track resolution of 70 cm, and we used the ALT13 products in the study (Rebold et al., 2021, https://icesat-2.gsfc.nasa.gov/science/data-products). All altimetry data are then referenced to the EGM2008 (Pavlis et al., 2012) geoid model. The source and version of each altimetry product are listed in Table 2.

**Table 2. Summary of altimetry datasets used in this study.**

| Satellite | Data period | Retrackers | Repeat cycle |
|---|---|---|---|
| CryoSat-2 | 2010.04-2021 | PPCOG, NPPTr[0.5], NPPTr[0.8] | 369 days |
| SARAL/AltiKa | 2013.02-2021 | ICE-1, ICE-2, PPCOG, NPPTr[0.5], NPPTr[0.8] | 35 days before 2016 July and subcycles of 15-17 days after 2016 |
| Sentinel-3A | 2016.02-2021 | OCOG, Ocean, PPCOG, NPPTr[0.5], NPPTr[0.8] | 27 days |
| Sentinel-3B | 2018.04-2021 | OCOG, Ocean, PPCOG, NPPTr[0.5], NPPTr[0.8] | 27 days |
| Jason-3 | 2016.02-2021 | OCOG, Ocean, Adapt | 10 days |
| ICESat-2 | 2018.09-2021 | Official | 90 days |

**Note: PPCOG refers to the Primary Peak Center Of Gravity algorithm. NPPTr[0.5] and NPPTr[0.8] refers to the Narrow Primary Peak Threshold retracker with a 50% and 80% threshold level algorithm, respectively. OCOG refers to the traditional Offset Center Of Gravity algorithm.**

The flowchart of constructing reservoir water level time series is provided in Appendix B (Figure B1). It contains three steps. Firstly, we picked up valid satellite altimetric measurements by selecting the correct ground tracks and valid footprints falling on reservoirs. This was done by masking the altimetry data from the GRanD polygons of reservoirs. Secondly, we constructed the reservoir point water level measurements via the following equations:

$$WSE = H_{alt} - R_{range} - N_{geo}, \qquad (1)$$

$$R_{range} = R_{trac} + R_{retrac} + R_{atm} + R_{geo}, \qquad (2)$$

where $H_{alt}$ refers to the altitude of satellite, $N_{geo}$ is the height of EGM2008 geoid, and $R_{range}$ is the range that measures the distance from water to satellite. $R_{trac}$ is the range to the nominal bin of the waveform and $R_{retrac}$ denotes the re-tracking correction. $R_{atm}$ and $R_{geo}$ are the atmospheric corrections (wet tropospheric, dry tropospheric, and ionospheric corrections) and the geophysical corrections (solid earth, pole, and ocean loading tides). These corrections are taken from their products. Thirdly, to construct reservoir water level time series, the following sub-steps are carried out:

- Altimetry-derived WSE are pre-selected based on the water occurrence map (occurrence > 10%, set 80% for CroSat-2) of the Global Surface Water Explorer (https://global-surface-water.appspot.com/).
- We removed outliers for each pass (i.e., 2 deviations away from the median value) using the median of absolute deviation (Jiang et al., 2019).
- Outliers are identified and discarded by comparing with SRTM DEM, i.e., 20 m away from DEM (set 40 m for reservoirs with large fluctuations).
- The remaining WSE measurements are applied to construct time series based on the R package "tsHydro" available from (https://github.com/cavios/tshydro). This package efficiently estimates along-track water level in the case of outlying measurements (Nielsen et al., 2015).

As a result of the above steps, we generated standard measurement (SM) reservoir water level time series products from each satellite altimetry with different retracking algorithms (Figure B1). In general, SM products have a relatively low resolution determined by the spatial sampling pattern and repetitive period of a satellite altimeter. For example, Sentinel-3A spaced its ground tracks 104 km apart at the equator, thus only potentially offering altimetric measurements for 194 GRanD reservoirs in China, while Cryosat-2 can visit 873 GRanD reservoirs in China but has a repeat period of 369 days. To cope with the limitation of the opposing spatial sampling and temporal resolution of single altimeter and obtain an enhanced resolution water level product, we merged single-satellite SM products from multisource (i.e., CryoSat-2, S3A, S3B, SARAL/AltiKa, ICESat-2, and Jason-3) for a reservoir if available and generated enhanced measurement (EM) reservoir water level time series products. Notably, we select the SM products from each satellite with the best retracking algorithm in terms of root-mean-square error (RMSE) compared to in situ water level or the default retracking algorithm time series to densify time series. To remove inter-satellite systematic biases, two approaches are used: the first one is applied to satellites with enough overlapping periods by directly removing their mean water level differences, and the second one is to use the remotely sensed reservoir

area time series as an anchor of biased time series to estimate the inter-satellite relevant bias. We used the Gauss–Helmert adjustment scheme to minimize the 2-D cost function in surface-area–water-level coordinates (Figure B1).

To evaluate the performance of both SM and EM altimetric products, we calculate the RMSE values against in situ water level. The RMSE is a standard error metric in this field and calculated by comparing water level anomalies between gages and satellites.

## 2.2 Surface area datasets

In this study, we applied the new algorithm developed by Donchyts et al. (2022) to leverage freely accessible Landsat and Sentinel-2 images to generate reservoir water area time series. The GEE code for this water mapping algorithm is available at https://github.com/global-water-watch/research-reservoir-water-dynamics and was applied individually to each reservoir and every satellite image intersecting a given reservoir to map accurate reservoir water. This algorithm can efficiently address several challenges associated with optical Landsat satellites, such as contamination from clouds and limitations of previous algorithms that reclassify contaminated pixels as water. Donchyts et al. (2022) demonstrated the algorithm's good performance in mapping reservoir water areas by comparing the areas with in situ water level/storage in 768 reservoirs of varying size and geographic regions. Here, we detail how this algorithm addresses the challenges from optical images and generates water area time series. First, we selected the cloudy satellite images that intersect with a given reservoir shapefile. Second, we used the global cloud frequency dataset (Wilson, 2016) to identify the cloudiest images that are fully covered by clouds and corrected the remaining images using the following steps. Third, we computed the NDWI (normalized difference water index) spectral water index. Fourth, we detected land/water edges based on the Canny edge detector algorithm (Donchyts et al., 2016) and defined sampling areas for pixels around the land/water edges. Fifth, we determined the optimal threshold based on the Otsu thresholding algorithm (Markert et al., 2020) using a sample of NDWI spectral index values within the region determined in the previous step to obtain a water mask. Next, we eliminated incorrectly detected water (water pixels detected as non-water) by sampling surface water occurrence along water edges and obtained the final gap-filled water mask by clipping surface water occurrence at a given occurrence value and combining it with the water mask. Lastly, reservoir water area time series from the final gap-filled water mask are filtered with a quantile-based temporal outlier filtering algorithm to remove the remaining errors. Detailed procedures and flowcharts can be found in Donchyts et al. (2022).

After these steps, we generated monthly reservoir water area time series for 338 reservoirs. To analyze the performance of our products, reservoir time series are compared with the in situ water level time series, the altimetric water level time series (SM and EM product, see Section 2.1), and two similar existing products from GRSAD (Global reservoir surface area dataset, Zhao and Gao, 2018) and RealSAT (Khandelwal et al., 2022, Table 1). The CC, rRMSE (relative RMSE), and rBIAS (relative bias) are used as indicators of data quality.

### 2.3 Reservoir storage variation estimation

Monthly reservoir storage variation estimation is based on two common approaches: one is to use water level and water areas
from satellite altimeters and images, while another one is to use imagery-based water areas and DEM (digital elevation model).
The core of these two approaches is to construct the A–E relationships from the overlapping records of water level and areas
or DEM. Here, we assume that the A–E relationships can be described by five hypsometric relationships (i.e., linear, power,
exponential, polynomial, and logarithmic relationships). Parameters of the relationships are derived by minimizing the residual
sum of squares (RSS) using an ordinary least squares (OLS) regression. The curves were compared based on their $R^2$ values
and the one with the best performance is served as the hypsometry relationship of the reservoir. For reservoirs with enough
overlapping water level and area records from satellites, we performed the following procedures (Figure B2).

- The monthly WSE was estimated by directly averaging all measurements within each month.
- We generate the scatterplot of monthly area and water level data pairs and eliminating errors in the scatterplot.
- Generating the A–E relationship through OLS approaches.
- Applying the derived relationship to estimate WSE from SWA for periods when WSE is unavailable and inverse the function to estimate SWA from WSE for periods when SWA is unavailable (e.g., the month with large contamination ratio).
- Using Eq. (3), monthly RWSC estimation are determined during 2010–2021.

$$\Delta V_t = \frac{1}{2}(WSE_t - WSE_{t-1}) \times (SWA_t + SWA_{t-1}), \tag{3}$$

Regarding the DEM-based approached, we generated the water area-level-storage model based on SRTM-90m DEM and
reservoir shapefile (Vu et al., 2022), and then calculated RWSC by combining imagery-based water areas and reconstructed
area-level-storage model (Figure B2). After these steps, two types of reservoir storage variations are contained in our product.
To assess the data quality, we use the RMSE, Pearson correlation coefficient (CC), and normalized root-mean-square error
(NRMSE) as indicators of data quality. The generated RWSC were compared with in situ observation of 93 reservoirs.

### 3 Results and discussion

#### 3.1 Data set description

In this study, we generated the remotely sensed reservoir datasets for 338 Chinese reservoirs, with a total of 470.6 km$^3$ storage
capacity (50% reservoir water capacity in China). The geographical distributions of these reservoirs are represented in Fig. 1
and a summarized information on the components of the datasets are shown in Table 3. By synthesizing information from
various data sources, the remotely sensed datasets (WSE, SWA, and RWSC) of 338 Chinese reservoirs were calculated during
2010–2021 and are publicly available at https://doi.org/10.5281/zenodo.7251283 (Shen et al., 2021). The files provided are:
(i) the reservoir shapefiles, (ii) the time series of SWA, WSE and RWSC, and (iii) a readme file. In the directory of 01_res_loc,

we provide two ESRI shapefiles (the location of 338 reservoirs and 93 reservoirs with in-situ observations for validation) and one Excel file of their associated attributes. In the directory of 02_res_wse, we provide the time series of reservoir water surface elevation in two modes (i.e., standard-Measurement and enhanced-Measurement), with their comprehensive evaluation reports and figures in PDF and Excel files. The standard-Measurement (SM) products are individual measurements from each satellite altimeter with different retracking algorithms, while the enhanced-Measurement (EM) products are the densified water level observations from multisource if available. In the directory of 03_res_swa, we provide reservoir monthly area time series. In the directory of 04_res_rwsc, we provide the time series of RWSC in two modes (i.e., DEM-based, and water area and water level from satellites) and A–E curves, with their comprehensive evaluation reports, regression statistics, and figures in PDFs and Excels. Different levels of data are provided in an easily readable file format, showing that our remotely sensed datasets have clear patterns and can capture seasonal filling and emptying of reservoirs very well. For more details, please refer to the following sections and supplementary materials.

**Table 3. Summary of the data provided in this study.**

| Category | | Number of reservoirs | Description |
|---|---|---|---|
| **01 res_loc** | | 338 | Two shapefiles (338 reservoirs and dams, and 93 validated reservoirs) and one excel files associated with reservoir attributes |
| **02 res_wse** | Standard Measurements (In total: 338 reservoirs) | 111 | From Sentinel-3A mission, 27-days, 2016-2021, with 5 retracking algorithms |
| | | 117 | From Sentinel-3B mission, 27-days, 2018-2021, with 5 retracking algorithms |
| | | 146 | From SARAL/AltiKa mission, 35-days, 2016-2021, with 5 retracking algorithms |
| | | 243 | From CryoSat-2 mission, 369-days, 2010-2021, with 3 retracking algorithms |
| | | 26 | From Jason-3 mission, 10-days, 2016-2021, with 3 retracking algorithms |
| | | 147 | From ICESat-2 mission, 90-days, 2019-2021, with 1 retracking algorithm |
| | Enhanced Measurements | 196 | Enhanced measurements (EM) product by merging SM products, from 2010-2021, sub-monthly or monthly |
| **03 res_swa** | | 338 | Monthly from 2010-2021 |
| **04 res_rwsc** | Satellite water level-area based | 335 | Monthly storage variation from 2010-2021 |
| | DEM-based | 266 | Monthly storage variation from 2010-2021 |
| **Readme file** | | | A detailed description of the generated products and references |

### 3.2 Reservoir water level product

We provided reservoir WSE time series in two modes: SM (standard measurement) and EM (enhanced measurement) products extracted from six satellite altimeters (i.e., Sentinel-3A (S3A), Sentinel-3B (S3B), SARAL/AltiKa (SAL), Cryosat-2 (CS2), Jason-3 (J3), and ICESat-2 (IC2)). In total, 921 reservoirs are visited by the six altimetry missions over China during CryoSat-

2 era, providing basic WSE information. After outlier's removal, time series construction and combination, and visual inspection, we finally retain 338 reservoirs that have enough valid measurements. Note that, most reservoirs are removed due to the insufficient altimetry data points rather than other reasons. Out of 338 reservoirs, most reservoirs are visited by two drifting altimeters (i.e., 243 and 146 reservoirs by Cryosat-2 (CS2) and SARAL/AltiKa), while Sentinel-3A (S3A), Sentinel-3B (S3B), and Jason-3 (J3) cover 111, 117, and 26 reservoirs, respectively (Fig. 1). To evaluate the data quality, we followed the normal practice in the field by comparing WSE anomalies between satellites and gages by removing the mean value due to the unknown local vertical datums. Due to the missing observations of most reservoirs with in-situ records for the period of May 2018 to July 2019, we evaluate reservoirs where the overlapped WSE observations between satellites and stages are larger than 8, resulting in a total of 74 reservoirs with an average of 20 data for validation (34 by S3A, 23 by S3B, 9 by SAL, 27 by CS2, 11 by J3, 26 by IC2, and 74 by EM product). Performance of remotely sensed results is considered moderate and even good based on visual inspection of time series, statistical assessment, and reported accuracies from previous publications (Villadsen et al., 2016). In the next two paragraphs, we will show the data availability of SM and EM products as well as how well these remotely sensed results are in good agreement with gauged records.

Fig. 2 shows performance of six altimeters in terms of RMSE of retracked WSE with different retracking algorithms. No significant difference is observed in terms of RMSE values among these retrackers although they are performing differently. In most cases, all retrackers retrieve WSE consistently. Interestingly, all retrackers consistently perform poorly for some reservoirs although the reservoir areas are relatively large (Figure S2). Appropriately, OCOG algorithm is more robust over the most reservoirs for Jason-3. It should be noted that when merging SM products for a reservoir, the observations from the retracker that has the smallest RMSE are applied.

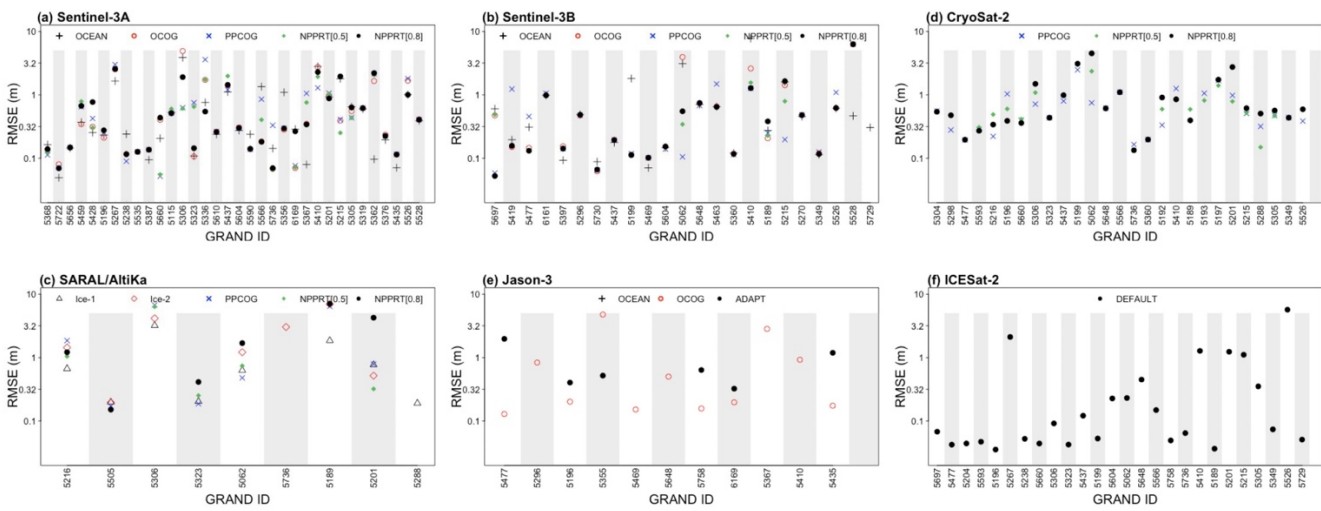

**Figure 2: Comparison of the different retracking algorithms (three for Cryosat-2 and Jason-3, and one for ICESat-2) of six altimeters at validated reservoirs. Logarithmic scales are used for Y-axis. X-label refers to the reservoir GRanD IDs. For some reservoirs occasionally, there is no useable data delivered by one specific retracking algorithm, and therefore no RMSE is available.**

Taking the observations from the retracker that has the smallest RMSE, we merge all observations for a specific reservoir from multisource if available. Fig. 3 shows the performance of merged WSE observations (i.e., EM products) in terms of RMSE values. Individually, the values of RMSE reveal that all altimetry missions can deliver useful water level measurements for reservoirs (Fig. 2). Specifically, percentages of reservoirs having very good RMSE values smaller than 0.3 m, moderate RMSE values ranging from 0.3 to 1.0 m, and relatively poor RMSE values over 1.0 m for each altimeter are 50%, 29%, 21% (S3A: validated in 34 reservoirs), 48%, 39%, 13% (S3B: 25), 38%, 37%, 25% (SAL: 9), 23%, 54%, 23% (CS2: 30), 55%, 27%, 18% (J3: 11), and 73%, 8%, 19% (IC2: 26), respectively. After merging observations from multisource if available, a total of 74 reservoirs are evaluated: 42 reservoirs have a good agreement with in situ data with a RMSE value below 0.6 m, among which 17 reservoirs have a very good data quality with a RMSE value smaller than 0.3 m. Another 29 reservoirs have a moderate RMSE value from 0.6 to 1.0 m. Around 4% has relatively poor performance in terms of RMSE values regardless of reservoir area. Some of them are located on the tributaries of the Yellow River and the Yangtze River. To demonstrate the data availability and reservoir water level time series provided in our SM and EM products, a selected number of example reservoirs presenting different areal size are shown in Fig. 4 and Fig. 5, respectively. All single-altimetric time series capture the dynamics of reservoir water level well, resulting in an improved temporal resolution of EM water level time series. For the remaining reservoirs with no in situ observations available, we give the SD (standard of error) estimates that quantify the accuracy of water level along the track at the level of individual data points. Detailed evaluation reports and PDFs representing water level time series for each reservoir are available in the datasets. By taking advantage of six satellite altimetry missions, we are able to densify WSE observations in most cases.

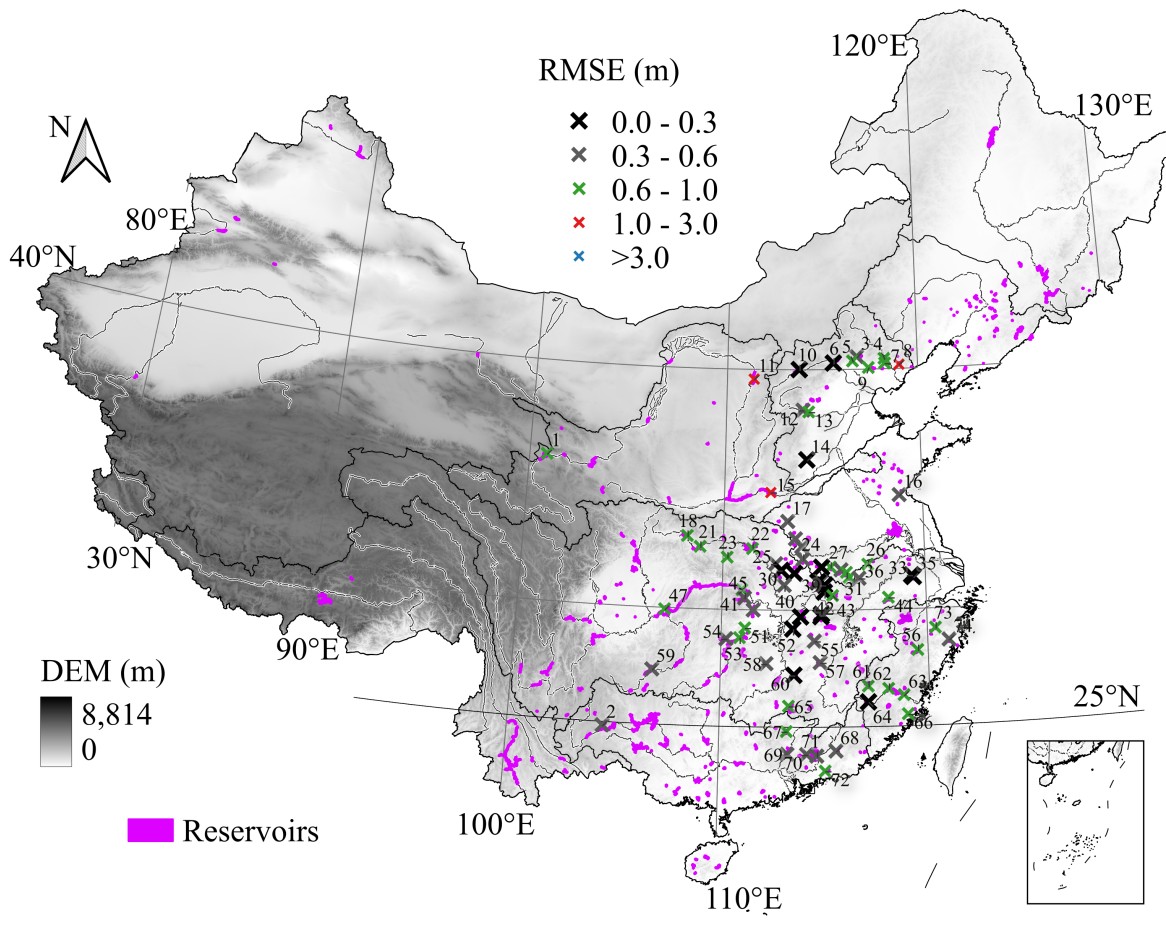

**Figure 3: Performance of the enhanced measurement products in terms of RMSE of 74 reservoirs. For validated RESERVOIR ID, please refers to the Supplementary.**

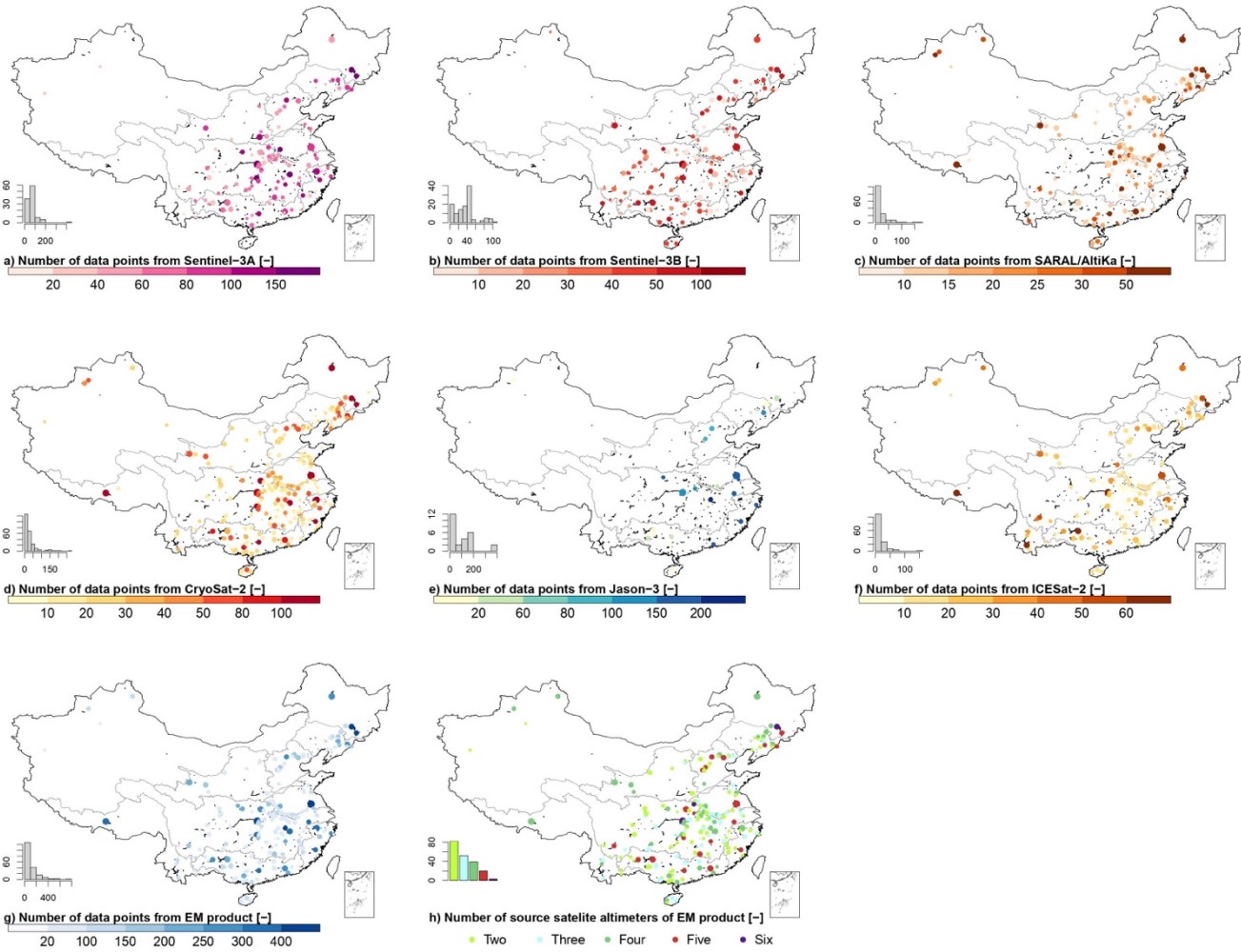

**Figure 4: The reservoir water level data availability of SM (standard measurement) and EM (enhanced measurement) products as well as the number satellite altimeters EM products.**

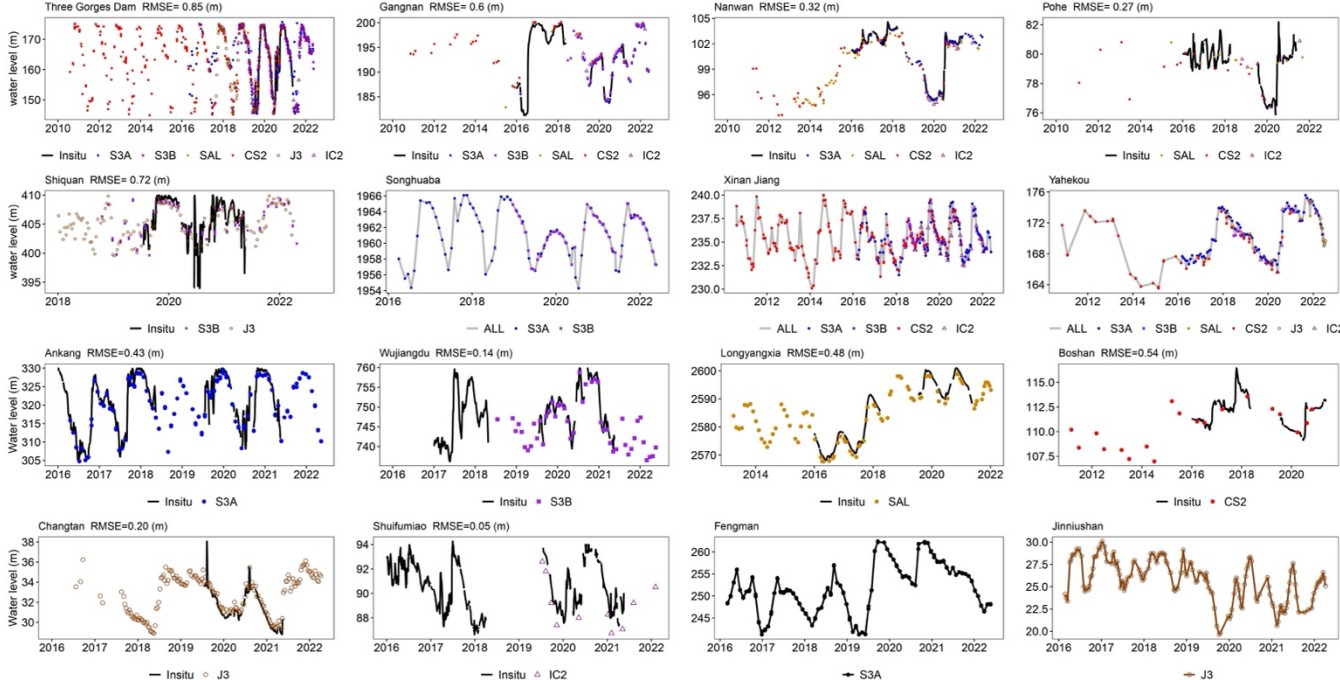

**Figure 5: Illustration of SM (standard measurement) and EM (enhanced measurement) reservoir water level time series at a selected sample of reservoirs with varying area size. Time series for other reservoirs are all available in the datasets.**

Note that the remotely sensed datasets developed by previous studies discussed in the Introduction are unavailable or do not match the coverage of any reservoirs in our dataset. Nonetheless, we compare 27 reservoirs of our dataset with three databases from G-REALM, DAHITI and Hydroweb and notice that our remotely sensed dataset is comparable (Fig. 6 and Fig. C1). If in-situ observations available, the time series from different sources are compared against in situ measurements and the RMSE values are calculated, otherwise; the CC values of WSE estimates of online database and our dataset are calculated. Across 7 gage comparisons (Fig. 6 d-j), G-REALM, DAHITI, Hydroweb and our dataset are similar and show close agreement with in-situ measurements. Nonetheless, there are some differences. For the Three Gorge, Xiaolangdi, and Shiquan reservoirs (Fig. 6 d-f), our dataset can be less noisy and better represent the dynamics in water level, with lower RMSE values than other sources. In case of the other four reservoirs (Fig. 6 g-j), the RMSE values of our dataset are slightly higher than those of Hydroweb, but still fall the satisfactory results below 0.60 m. It is worth noting that the time series of our dataset are much denser than those from Hydroweb and show clearer patterns (Fig. C1). The results of 21 reservoirs without in-situ observations indicate that all time-series show dynamics of reservoir water level very well, highlighting the critical contribution of both existing and our datasets. In most cases, our dataset shows good agreements with measurements from others, with CC values > 0.9. Nonetheless, there are some differences. Systematic biases are in these databases for the geoid issue (Fig. 6 j). In addition, some large discrepancies can be found in certain reservoirs, e.g., the Shuifeng reservoir (Fig. 6 c) did not show a clear fluctuation pattern as captured by G-REALM; the periods in 2020 between our dataset and Hydroweb at the Fengman reservoir (Fig. 6 b). Our datasets are denser than Hydroweb over most reservoirs (Fig. C1) and can be less noisy. These advantages

would benefit the continuity and accuracy of the remotely sensed WSE and RWSC. Overall, this section demonstrated that performance of our datasets approximates accuracy of existing global altimetry datasets.

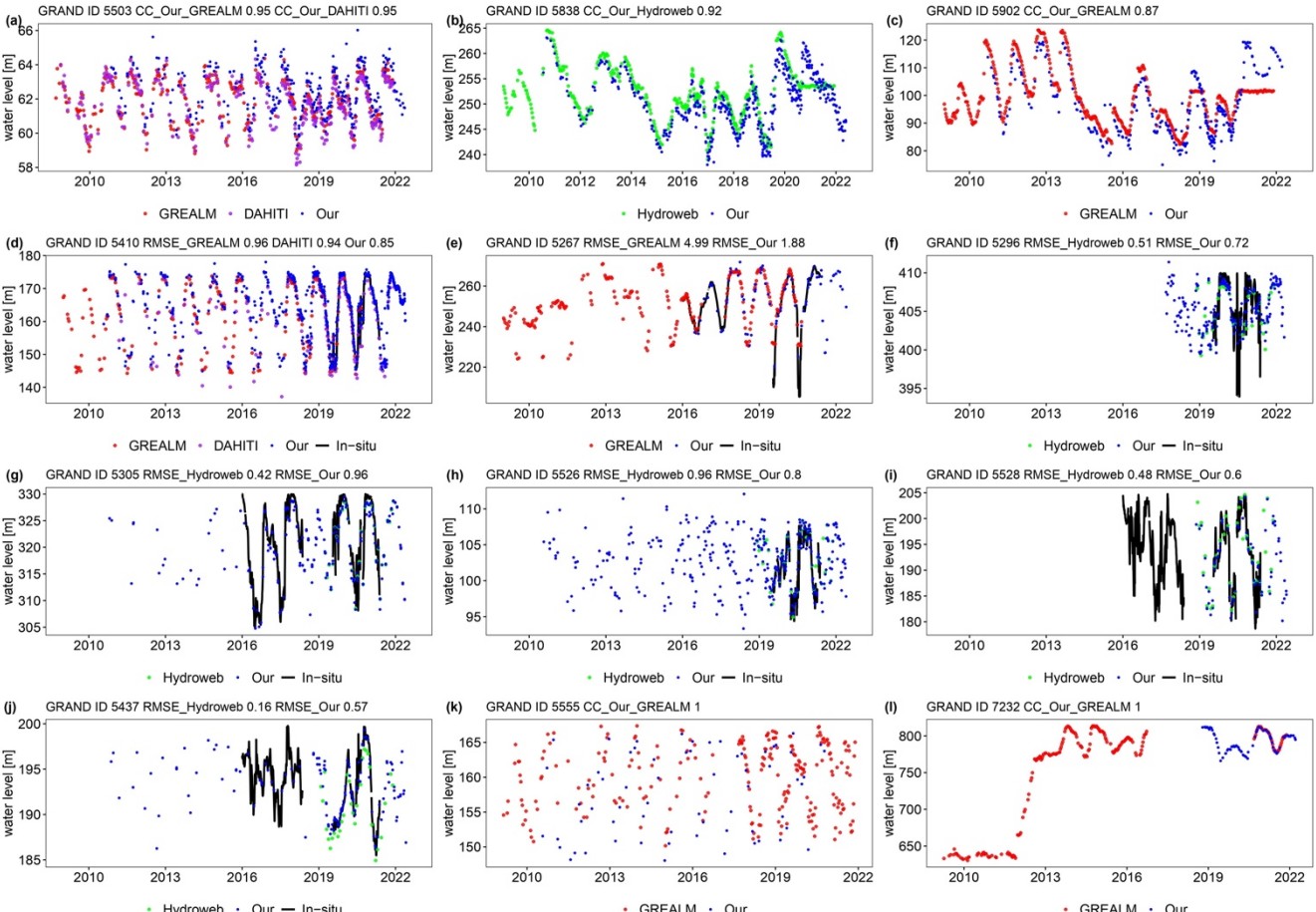

**Figure 6: Multiproduct evaluation at p27 reservoirs (15 reservoirs are shown in Fig. C1). DAHITI is plotted in black, G-REALM in red, Hydroweb in green, our dataset in blue, and in-situ records in black line. RMSE values are given when in-situ observations are available, otherwise, CC values are given at the top of each subplot. (a)-(l) are the Zhelin, Longyangxia, Fengman, 5902, Three Gorge, Xiaolangdi, Shiquan, Ankang, Wuqiangxi, Fengtan, Geheyan, Zhexi, Nuozhadu reservoir, respectively.**

### 3.2 Reservoir water area product

Monthly reservoir SWA time series are provided for 338 reservoirs during 2010-2021 and are compared with water level time series (in situ and altimetric measurements) and two other similar areal datasets water level. The SWA time series show good agreements with in situ water level observations in 93 reservoirs, approximately 80% have good CC values exceeding 0.5, among which 48 reservoirs show very good agreement with a CC > 0.8. Comparing to our altimetric standard measurements, we found that reservoir SWA and altimetric products generally show a good agreement with CC values higher than 0.5 for 70%

of 323 validated reservoirs, among which 139 reservoirs have very good agreement with a CC value > 0.8. Comparing to our altimetric enhanced measurements, reservoir SWA and altimetric products also show a good agreement with CC values higher

than 0.5 for 73% of 196 validated reservoirs, among which 62 reservoirs show very good agreement with a CC value > 0.8. In addition, two similar areal datasets (Table 1), i.e., GRSAD (Zhao and Gao, 2018) and RealSAT (Khandelwal et al., 2022), were used for cross validation. GRSAD provides monthly SWA values for global 7,246 reservoirs during 1984-2020 (updated version 3) extracted from the Landsat-based images (Pekel et al., 2016) and correction of contaminations from terrain shadows, clouds, and cloud shadows. The datasets were validated over 9 reservoirs with in situ water level/storage observations and compared with the synthetic data from cloud-clear Landsat images, showing a good performance of the algorithm to repair contaminated optical images for more reliable SWA estimates. RealSAT used a machine-learning method (i.e., ordering based information transfer) to process optical images for generating monthly SWA values over 681,137 global lakes/reservoirs from 1984 to 2015. It should be noted that GRSAD used the existing reservoir shapefiles from GRanD database to generate SWA values, while RealSAT generated new lake polygons from surface water occurrence data. Based on all compared reservoirs available, we found that our SWA time series show good agreements to values in GRSAD (median CC value of 0.64, rBIAS = -9%, rRMSE = 26%, n = 338) and RealSAT (median CC value = 0.68, rBIAS = -10%, rRMSE = 22%, n = 47) datasets. Overall, these comparisons (Fig. 7) above suggest a good level of trustworthiness in our SWA time series.

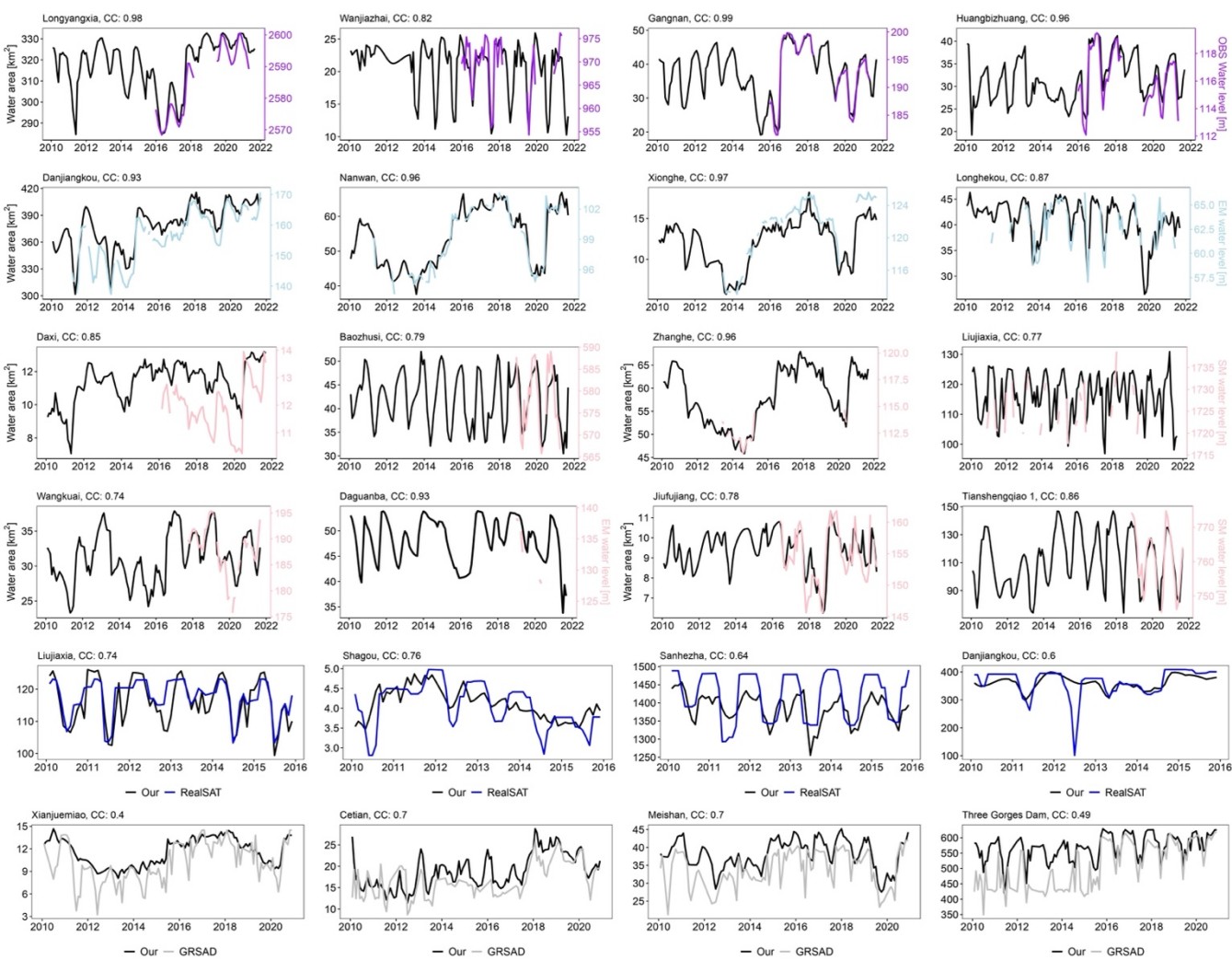

**Figure 7: Illustration of reservoir water area time series against in situ water level, altimetric water level from standard measurement and enhanced measurement, as well as GRSAD and RealSAT area time series at a selected sample of reservoirs with varying area size. Time series for other reservoirs are all available in the datasets.**

### 3.3 Reservoir storage variation product

We provided monthly RWSC time series from 2010 to 2021 in two modes: one is to use WSE and SWA from satellite altimeters and images, while another one is to use satellite SWA and area-storage model developed by DEM. After excluding reservoirs without insufficient WSE-SWA data pairs to establish the A–E relationships and visual inspection of time series, we finally retained 337 reservoirs with RWSC time series, among which 335 reservoirs have RWSC estimates derived from the first type method while 266 reservoirs have RWSC estimates derived from the DEM-based method. To evaluate the data quality, we

compared with in situ storage data of 91 reservoirs and calculated three error statistical metrics (i.e., RMSE, normalized root-mean-square error (NRMSE), and CC). The A–E curves derived from satellite WSE and SWA data are evaluated based on

their $R^2$ values. We notice that 69% reservoirs of A–E curves could be better explained by a second-order polynomial function, while 13% and 16% reservoirs of A–E relationships are assumed to give a power and exponential function (Fig. 8 f, h). A total of 283 of 335 reservoirs (84%) have moderate $R^2$ values > 0.5, among which 107 reservoirs show very good agreement with

$R^2$ values > 0.8. Nevertheless, 15% has relatively poor performance in terms of $R^2$ values. Overall, our A–E curves are reliable and lay the good foundation for RWSC estimates. Across gauge comparisons of RWSC, the median statistics of CC, NRMSE, and RMSE are 0.89, 11%, and 0.021 $km^3$. Around 91% reservoirs (83 of 91) show good data quality with a NRMSE value below 20% and a RMSE value ranging from 0.002 to 0.31 km3. The lowest NRMSE is 4%, from the Gangnan reservoir that displays high CC and low RMSE values. Regarding the DEM-based RWSC estimates, the results are getting worse, with the

median statistics of CC, NRMSE, and RMSE are 0.56, 20%, and 0.03 $km^3$. The errors can be attributed to the inaccuracy of the area-storage model developed by DEM. It should be noted that this type of RWSC estimates is served as an alternative product. Figure 8 shows examples of RWSC for some selected small, medium, and large reservoirs located in different climate zones. Closer examination in Fig. 9 seems to indicate that almost all remotely sensed RWSC estimates show similar patterns to the observations, i.e., both positive or negative, despite of some large discrepancies when capturing peak values. Nonetheless,

there are some differences. Some reservoirs with good NRMSE and RMSE values show poor performance in terms of CC value, e.g., the Baiguishan reservoir (CC: 0.38, NRMSE: 16%, RMSE 0.03 $km^3$) that experiences relatively significant surface water dynamics. Moderately poor performance of 20 reservoirs (7%) in terms of high NRMSE/RMSE and low CC values (CC < 0.4) is likely associated with their poor performances from the remotely sensed WSE and SWA. Overall, we used in-situ observations of 91 reservoirs as an important reference to validate RWSC dataset, thus bringing the good level of confidence

in our data quality.

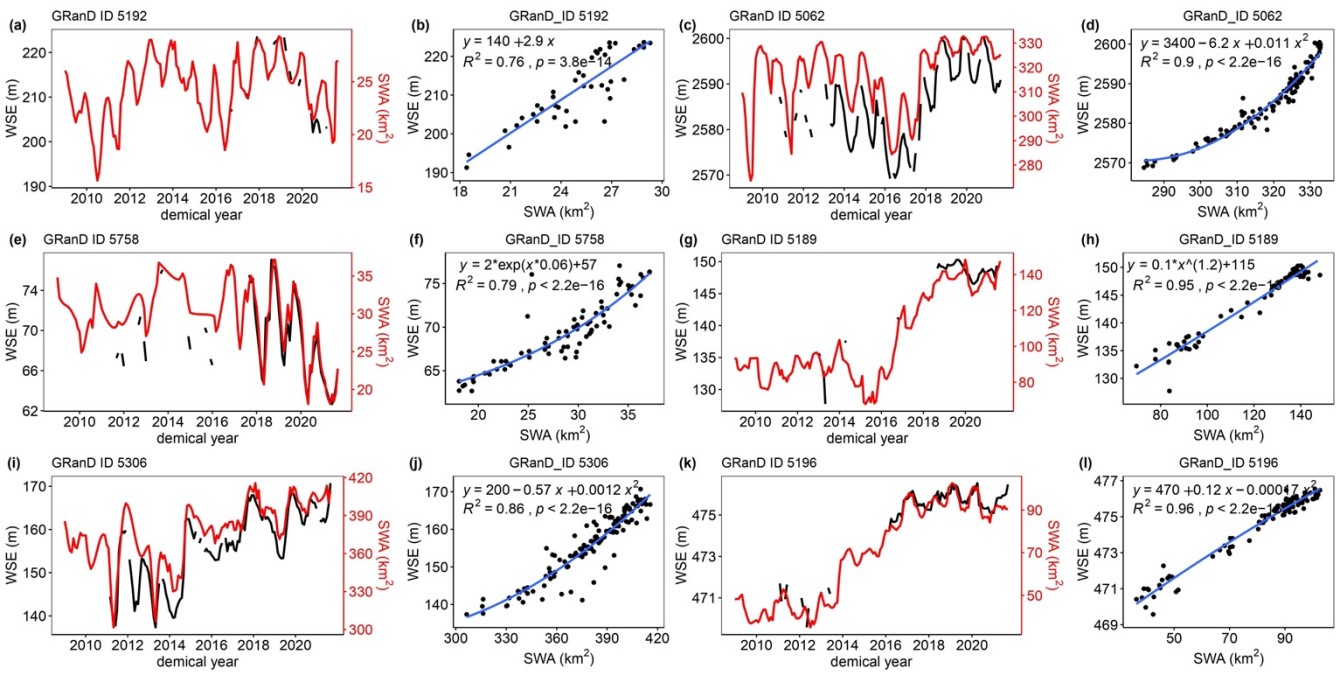

**Figure 8: Illustration of A–E relationships constructed by satellite WSE and SWA and their associated time series at six reservoirs. (a)-(k) are the Panjiakou, Longyangxia, Baipenzhu, Miyun, Danjiangkou, and Guanting reservoir, respectively. Note that: time series of WSE and SWA and associated established A–E relationships of the remaining reservoirs are available in our datasets.**

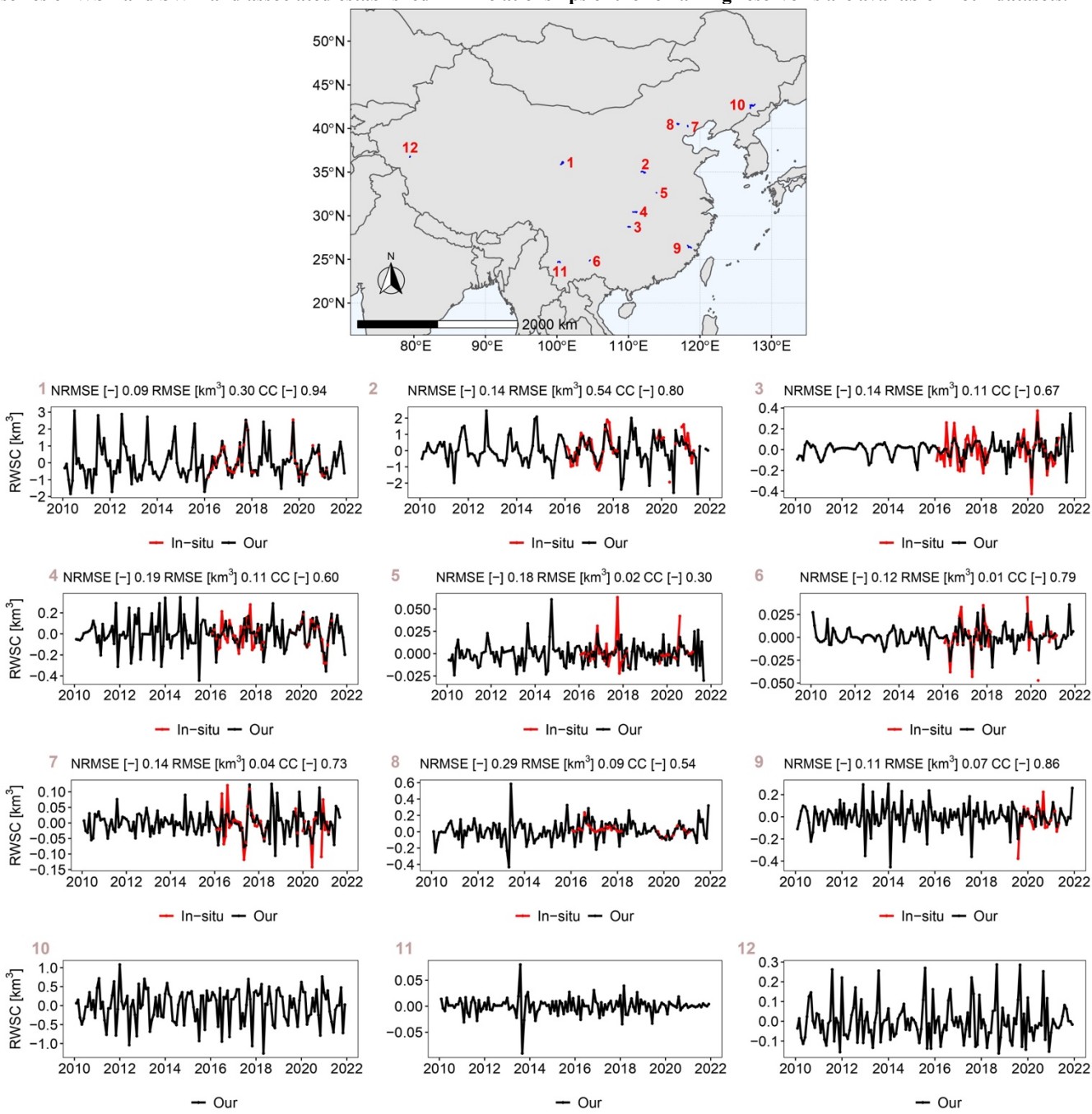

**Figure 9: Illustration of time series of the remotely sensed RWSC of 12 reservoirs. NRMSE, RMSE (km³), and CC values (if available) are given at the top of each subplot. Time series of the remotely sensed RWSC of the remaining reservoirs (validated or not validated) are available in our datasets.**

## 4 Applications

As explained earlier, our motivation is to develop the remotely-sensed reservoir datasets that can be applied as constraints to calibrate models or directly used for reservoir studies (Yigzaw et al., 2018; Shin et al., 2019, 2020). One of the most interesting scientific work that can be done with our datasets is to estimate how hydrographs in China have changed because of reservoir regulation. In order to do that, we need to combine inflow modeling with reservoir storage changes to estimate reservoir release. Fig. 10 demonstrates the flowchart of combining the process-based models or lumped models with our remotely-sensed RWSC datasets to achieve this goal. Recent studies started adopting this framework to assess the effect of dams and reservoirs on streamflow regulation, and/or downstream flood inundation (Gutenson et al., 2020; Zhong et al., 2020; Tavakoly et al., 2021). Here, following the normal but simple practices (Bonnema and Hossain 2019; Han et al., 2020), we estimate reservoir release using our remotely-sensed RWSC dataset and inflow simulated by a calibrated lumped hydrological model (i.e., GR4J, Génie Rural à 4 paramètres Journalier model), to demonstrate the potential of our datasets to help achieve this goal (Fig. 10). This experiment is carried out at the Ankang reservoir, which has a water capacity of 2.58 $km^3$ and a water extent of 58 $km^2$, located in the Han River. The basin-averaged precipitation from high quality GPM-Final products and potential evaporation are used to run the model. The Oudin approach is used to calculate the potential evaporation, and requires temperature from ERA5-Land products for calculation (Oudin et al., 2005). The model is pre-calibrated based on 10-year historical reservoir inflows (2001-2007 for calibration, 2008-2010 for validation). The Shuffled Complex Evolution (SCE-UA) is employed to calibrate the hydrological model through maximizing the Kling-Gupta Efficiency (KGE) value. Then, we simulate reservoir inflow during 2010-2020 in combination with our RWSC for release estimates. The values of KGE and streamflow hydrographs reveal that model performs well with KGE > 0.68 during both calibration and validation periods. The simulated releases show good agreements to the observations, with KGE exceeding 0.90 and NRMSE below 0.04. We compare reservoir inflow and release simulations and notice that flow regimes at the Ankang reservoir have been substantially altered (Fig. 11 f). In conclusion, our RWSC dataset can be applied to reservoir release simulation, achieving satisfactory streamflow simulations. However, some limitations can be seen in our case study. Firstly, reservoir evaporation and precipitation are neglected for the tested reservoir with humid climate conditions. We suggest that these variables should be considered using high quality satellite datasets such as ET products or model simulations. Secondly, the case study cannot provide a big picture of reservoir regulations on streamflow at national scale. Similar studies should be done at the remaining reservoirs to achieve this goal. Acknowledging such limitations, we argue that the datasets could help achieve the blueprint application by introducing the key components (e.g., RWSC) of reservoirs at national scale.

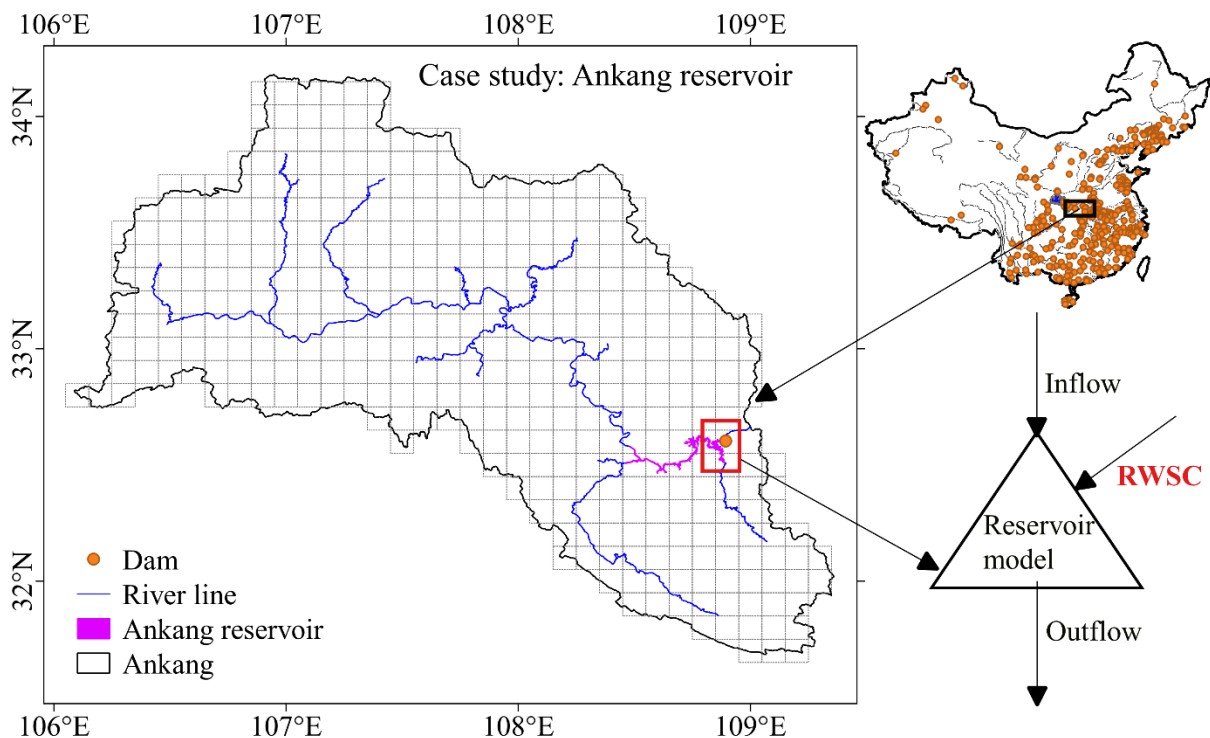

**Figure 10: Schematic representation of integration between models and our datasets for reservoir release application. Normally, streamflow at the node (i.e., the dam) should be replaced with regulated flow (i.e., reservoir outflow) and routed downstream by a routing model such as RAPID model. The remotely-sensed RWSC and inflow simulated by hydrological models are introduced to the reservoir model, i.e., the mass balance equation.**

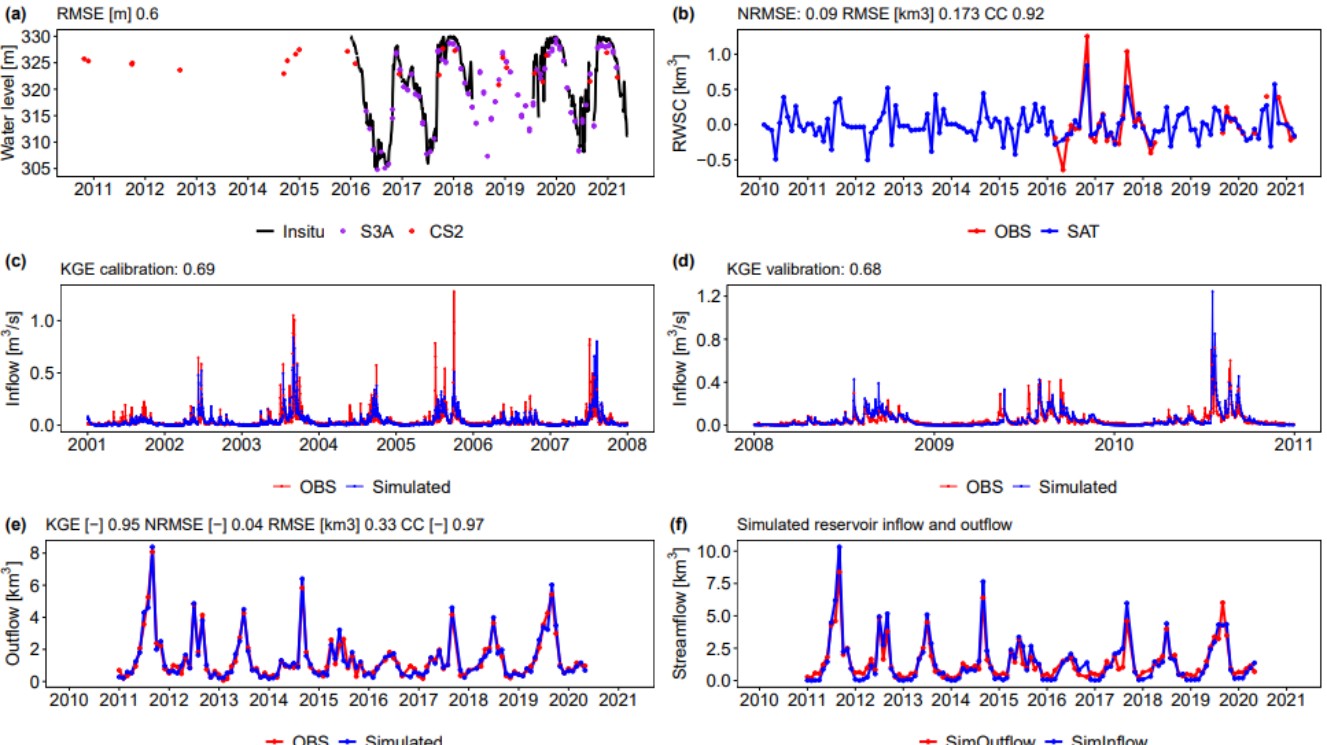

**Figure 11: Case study at the Ankang reservoir. (a) and (b) are the evaluation of the remotely-sensed WSE and RWSC. (c) and (d) denote the streamflow hydrographs simulated by GR4J model during calibration and validation periods. (e) represents the comparison of model simulated outflows and gauged records. (f) shows streamflow regulation by reservoirs. Note that: only historical inflow records before 2010 are available in this study.**

The datasets can benefit other applications across multiple disciplines in addition to areas described above. We highlight three areas for future applications. First, the RWSC can be used to develop a reservoir storage forecast system (Tiwari and Mishra, 2019) at 1- to 3-month lead that can be valuable for water resource management in China. Second, the datasets can be joined with hydrological and climate datasets to synthesize changes in water quantity and quality. For example, the datasets could be combined with carbon dioxide emissions from Carbon Monitor CHINA (https://cn.carbonmonitor.org/) to address questions that how changes in reservoir storage may co-vary with changes in carbon dioxide emissions. Third, the datasets can be extended to include other countries and thousands of small reservoirs, in the background of booming satellites such as the Surface Water and Ocean Topography mission that detects smaller water bodies (Biancamaria et al., 2016).

## 5 Conclusions

In this study, we utilize six satellite altimetry missions from SARAL/AltiKa, Sentinel-3 A/B, CroySat-2, Jason-3, and ICESat-2 in combination with water area data from Landsat and Sentinel-2 images, to develop high-resolution reservoir datasets of WSE, SWA, and RWSC. The resulting datasets include 338 reservoirs with a total of 470.6 km$^3$ water storage accounting for 50% reservoir capacity in China. The remotely-sensed results are validated against gauged measurements of 93 reservoirs: (1)

The comparisons indicate the relatively high reliability and accuracy of monthly RWSC estimations, with 91% reservoirs (83 of 91) having good RMSE values from 0.002 $km^3$ to 0.31 $km^3$ and NRMSE values < 20%. For RWSC, the median statistics of CC, NRMSE, and RMSE are 0.89, 11%, and 0.021 $km^3$. (2) Satisfactory results and good agreements can be found between the WSE retracked by six altimeters and gauges. Individually, the percentages of reservoirs having good data quality with RMSE values below 0.3 m, moderate RMSE values from 0.3 to 1.0 m, and relatively poor RMSE values over 1.0 m for each altimeter are 50%, 29%, 21% (S3A: validated in 34 reservoirs), 48%, 39%, 13% (S3B: 25), 38%, 37%, 25% (SARAL/AltiKa: 9), 23%, 54%, 23% (CS2: 30), 55%, 27%, 18% (Jason-3: 11), and 73%, 8%, 19% (ICESat-2: 26), respectively. After merging WSE observations from multisource if available, a total of 73 of 74 (96%) reservoirs have good and moderate data quality with a RMSE value below 1.0 m, among which 42 reservoirs show good RMSE values below 0.6 m and 17 reservoirs show very good RMSE values < 0.3 m. By taking advantage of four missions, we are able to densify WSE observations in most cases. Nonetheless, for reservoirs accounting the remaining 50% water storage capacity, current satellite altimetry missions are not able deliver enough useful observations or detect these reservoirs given by their sparse altimetric ground tracks, therefore, not included in our products. Developing more general algorithms with better performance regardless of the reservoir's attributes and using satellite altimetric data with higher temporal resolution (e.g., Surface Water and Ocean Topography mission, Biancamaria et al., 2016) will be our next studies. Overall, our study fills such a data gap by incorporating various satellites into a comprehensive reservoir data set at national scale. We envision this dataset can be immediately applied to some scientific areas described in Sect. 4 and can provides strong support for many aspects such as hydrological processes and water management studies.

## 6 Data availability

All the generated remotely-sensed reservoir datasets are archived and available at https://doi.org/10.5281/zenodo.7251283 (Shen et al., 2021). They are distributed with a CC-BY license.

**Supplements.**

The supplement related to this article is available online.

## Appendix A.

 **Table A1. Providers of water level (upper panel), area (middle panel), and storage variation (bottom panel) time series for Chinese reservoirs.**

| Data sources | No. of reservoirs | Time and temporal resolution | Download link |
|---|---|---|---|
| **Hydroweb** | 32 | 1992–2021, 10–35 day | http://hydroweb.theia-land.fr/ |
| **DAHITI** | 8 | 2002–2021, 10–35 day | https://dahiti.dgfi.tum.de/en/ |
| **G-REALM** | ~30 | 1992–2021, 10–35 day | https://ipad.fas.usda.gov/cropexplorer/global_reservoir |
| **Tortini et al. (2020)** | <10 | 1992–2018, sub-monthly | https://doi.org/10.5067/UCLRS-GREV2 |
| **Shen et al. (2021)** | 338 | 2010–2021, monthly | https://doi.org/10.5281/zenodo.7251283 |
| **Bluedot** | not clear | 2016–2021, sub-monthly | https://blue-dot-observatory.com/ |
| **GRASD** | 923 | 1984-2018, monthly | https://doi.org/10.18738/T8/DF80WG |
| **Tortini et al. (2020)** | <10 | 1992–2018, sub-monthly | https://doi.org/10.5067/UCLRS-AREV2 |
| **RealSAT** | 85,522 (lakes and reservoirs) | 1984–2015, monthly | https://doi.org/10.5281/zenodo.4118463 |
| **Donchyts et al. (2022)** | 9,418 | 1985–2021, monthly | https://doi.org/10.6084/m9.figshare.20359860 |
| **Yao et al. (2019)** | ~8 | 1992–2018, sub-monthly | https://lakewatch.users.earthengine.app/view/glats |
| **Shen et al. (2021)** | 338 | 2010–2021, monthly | https://doi.org/10.5281/zenodo.7251283 |
| **Vu et al. (2022)** | 10 | 2008–2020, monthly | https://doi.org/10.5281/zenodo.6299041 |
| **Hou et al. (2022)** | 923 | 1984–2015, monthly | Not publicly accessible |
| **Tortini et al. (2020)** | <10 | 1992–2018, sub-monthly | https://doi.org/10.5067/UCLRS-STOV2 |
| **Shen et al. (2021)** | 337 | 2010–2021, monthly | https://doi.org/10.5281/zenodo.7251283 |

## Appendix B. The flowcharts of the proposed method for generating reservoir water level and storage variation products.

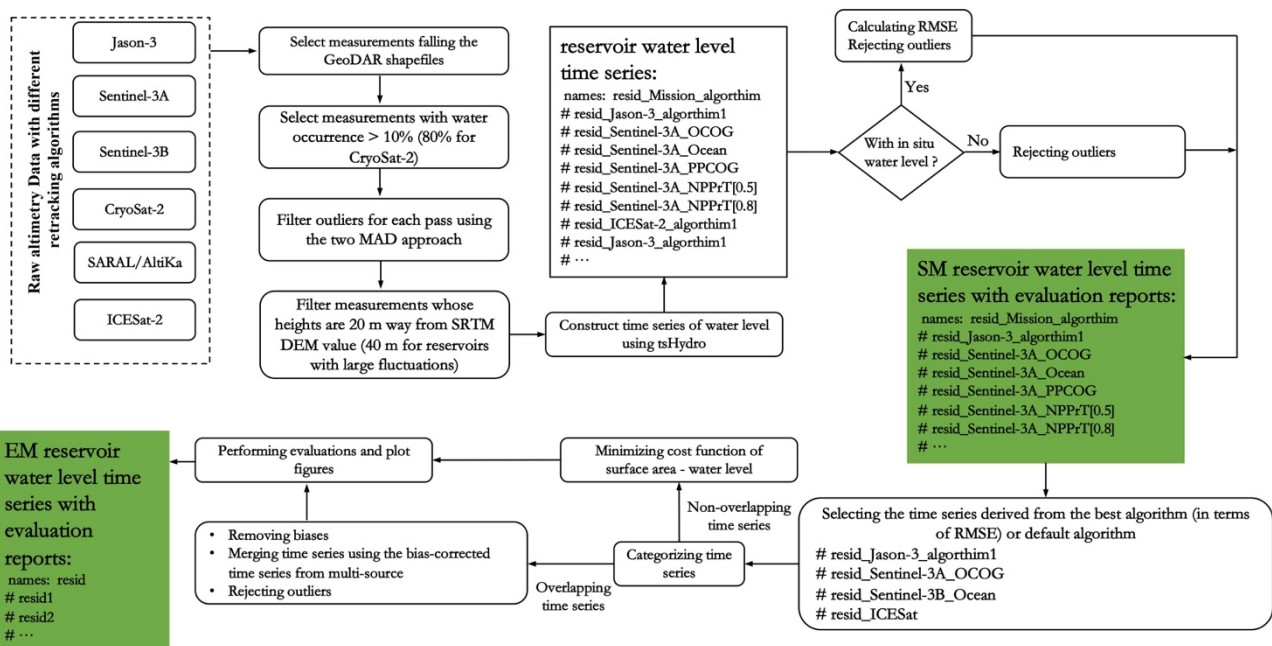

**Figure B1. Flowchart of obtaining SM (standard measurement) and EM (enhanced measurement) altimetric water level time series over reservoirs.**

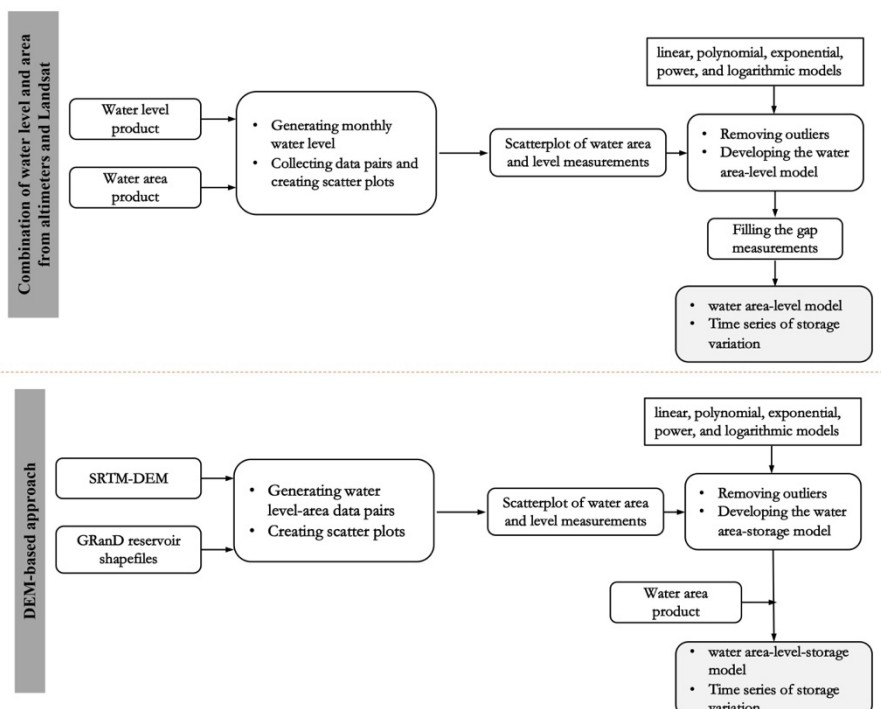

**Figure B2. Flowchart of obtaining water storage variation using reservoir water area from satellite imagery and water level from satellite altimetry (top panel). Flowchart of obtaining water storage variation using surface water area from satellite imagery and SRTM-DEM (bottom panel).**

**Appendix C.**

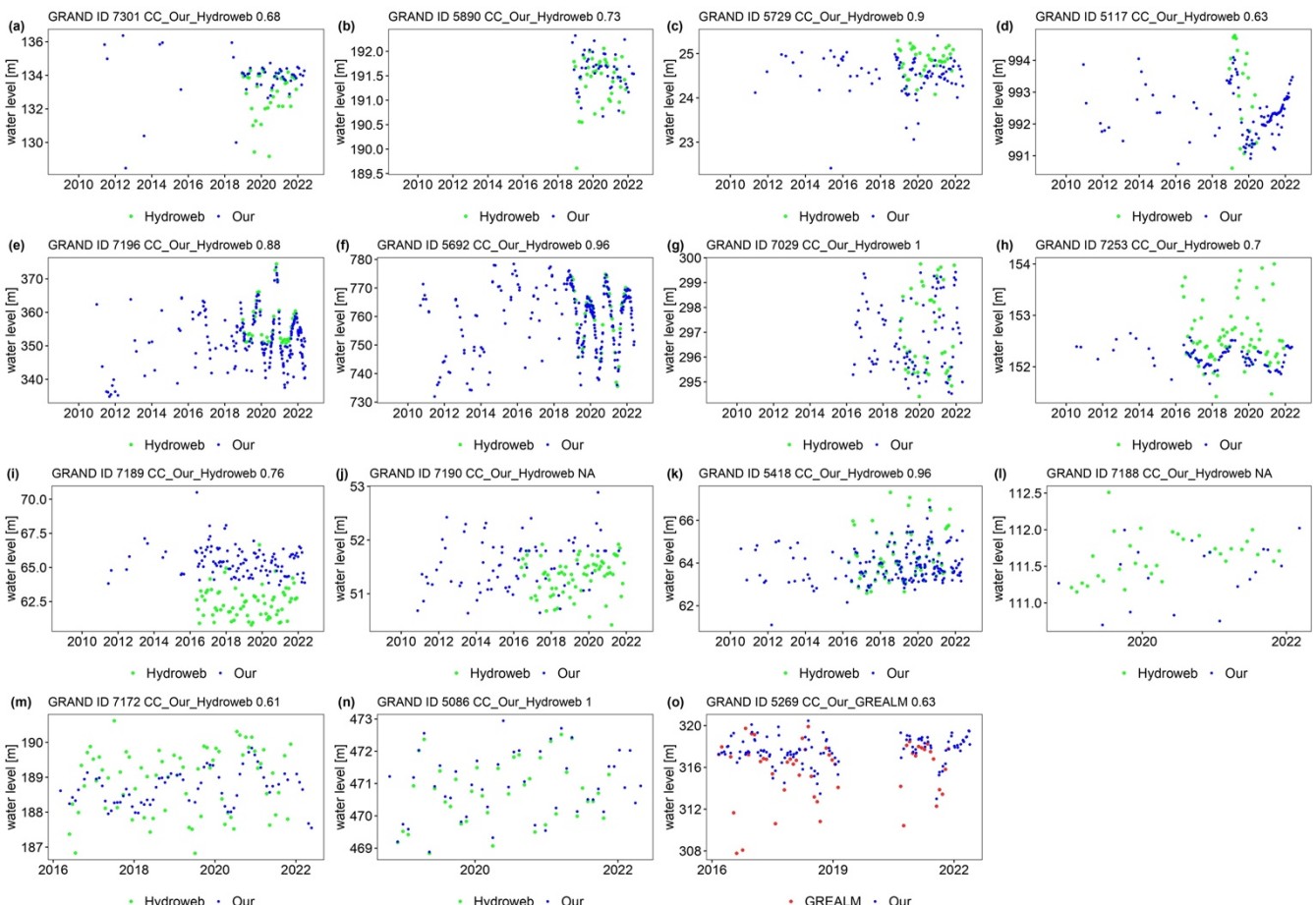

**Fig. C1: Multiproduct evaluation at 15 reservoirs. DAHITI is plotted in black, G-REALM in red, Hydroweb in green, our dataset in blue, and in-situ records in black line. RMSE values are given when in-situ observations are available, otherwise, CC values are given at the top of each subplot.**

**Author contributions.**

Y.S. and D.L. initiated the investigation. Y.S., D.L., L.J., and J.Y. designed the research. Y.S. processed the data and created the figures. K.N. initially extracted and processed the altimetry data. Y.S. prepared the manuscript with contributions from all co-authors.

**Competing interests.**

The authors declare that they have no conflict of interest.

**Acknowledgements.**

The authors acknowledge following data centers for providing original data:

- CryoSat-2 data: The baseline C level 1b dataset are from ESA (https://science-pds.cryosat.esa.int/)
- SARAL/AltiKa, and Jason-3 data from CNES AVISO+ (ftp://avisoftp.cnes.fr/AVISO/pub/)
- Reservoir and dam data from the GRanD (http://globaldamwatch.org/grand/) database
- Sentinel-3 level 2 data from Copernicus Open Access Hub (https://scihub.copernicus.eu/dhus/)
- Daily water level and storage data for 93 reservoirs from the local watershed agency (http://xxfb.mwr.cn/index.html) and National Hydrological Information Centre (http://113.57.190.228:8001/web/Report/BigMSKReport)
- ICESat-2 ALT08 products (https://nsidc.org/data/atl08/)

**Financial supports.**

This research has been funded by the National Natural Science Foundation of China (grant nos. 51579183 and 51879194) and the Danida Fellowship Centre (grant no. 18-M01-DTU).

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
