# Peer review of "High-resolution water level and storage variation datasets for 338 reservoirs in China during 2010–2021"

_Earth System Science Data, 2021_

## Community Comment (CC1)

Dear authors,

thank you for submitting such an interesting paper which is likely to provide a very valuable data source for water resources studies in China. As a potential end user of this dataset, I have a suggestion for the authors to help further improve the quality of the dataset.

The figure below depicts the RWSC of GranD ID 5062, 5267 and 5410 provided by the authors in Zenodo and the in-situ RWSC collected by myself. While the authors' data generally match with the observations, several abnormal spikes (outliners) exist for these dams, for example in mid-2011 and mid-2012 for GRanD 5062, and mid-2010 and late 2013 for GRanD 5410. These spikes could lead to an overestimation of CC reported by the authors, because these spikes seem to occur less in late 2010s where the validation is performed. From my point of view, this may be a result of contaminated data points from either altimetric satellites or remote sensing images. Therefore, I would like to encourage authors to conduct a rigorous data quality control to remove any possible problematic data points before publishing the final dataset.

[Figure]

**Figure 1.** RWSC of three dams provided by authors compared with the in-situ RWSC.

---

## Author Comment (AC2)

**Reviewer #1 Comment on essd-2021-470 (Stefano Galelli)**

Dear Stefano Galelli,

Thank you for your time and efforts in reviewing our manuscript. We are very happy to hear your positive feedback on our datasets which provide strong support for many aspects. We are very happy about the agreement that the presented data set is an important contribution to the large-scale studies that will take place in the context of increasing activity in the reservoir and dam sector and are pleased to be able to contribute with this data set. Please find attached point-to-point responses regarding your comments (marked in **purple**) and made corresponding changes in the main manuscript (in **red**). We hope that the improved manuscript can help the readers to better understand our study.

Kind regards.

**General comments**

Manuscript essd-2021-470 describes a novel dataset providing water surface, level, and storage information for 338 reservoirs in China. In my opinion, this is a much-needed dataset that fills in an important gap, since data on water reservoirs are typically not available to the international community. I believe many studies and downstream applications will thus benefit from these data.

R1C0: Thank you for your positive comments identifying the strengths of our work.

Overall, both manuscript and dataset are well organized, although a few important probably deserve more attention. In particular:

1. I am not entirely convinced about the approach used to estimate the hypsometric relationships, which, if I understand correctly, are based on water level and surface data estimated from satellite data. In general, water level data are rather reliable, while it is always a challenge to get the right water surface data (a matter that explains the use of image enhancing techniques), a problem that might affect the quality of the curves. So, why not using a DEM to get the right curves? This could be done for many reservoirs. Estimating the hypsometric relationships from a DEM would also limit the need for water surface data.

R1C2: Yes, you are correct! In this study, we constructed hypsometry curve (SWA-WSE curve) for each reservoir using satellite-based water level and surface water area datasets. We then apply this relationship to estimate WSE from SWA for periods when WSE is unavailable and inverse the function to estimate SWA from WSE for periods when SWA is unavailable. Using Eq.3, we calculated monthly reservoir water storage change (RWSC).

$$\Delta V_t = \frac{1}{2} (WSE_t - WSE_{t-1}) \times (SWA_t + SWA_{t-1}), \tag{3}$$

**Please note that we provided storage variation (i.e., RWSC), not storage**. To ensure the accuracy of hypsometric curves, we strictly select water level-area data pairs, i.e., "The monthly WSE was estimated by directly averaging all measurements within each month. Attributed to the denser and more frequent records of SWA, we selected SWA values with contaminations ratio smaller than 5% for the construction of hypsometry curve for each reservoir." The data pairs were assumed to give five hypsometric relationships, among

which the best one with highest R2 value is served as the hypsometry relationship of the reservoir. The derived RWSC results are evaluated against in-situ observations, based on our evaluations, favoring the success of the framework and our datasets. We give some references adopted the same methodology (i.e., constructing hypsometry relationships using satellite water level and area, then calculating RWSC) below.

**References:**

Gao, H., Birkett, C., and Lettenmaier, D. P.: Global monitoring of large reservoir storage from satellite remote sensing, Water Resour. Res., 48, W09504, doi: 10.1029/2012WR012063, 2012.

Bonnema, M., and Hossain, F.: Assessing the potential of the Surface Water and Ocean Topography Mission for reservoir monitoring in the Mekong River basin, Water Resour. Res., 55, 444–461, doi: 10.1029/2018WR023743, 2019.

Busker, T., de Roo, A., Gelati, E., Schwatke, C., Adamovic, M., Bisselink, B., Pekel, J.-F., and Cottam, A.: A global lake and reservoir volume analysis using a surface water dataset and satellite altimetry, Hydrol. Earth Syst. Sci., 23, 669–690, doi: 10.5194/hess-23-669-2019, 2019.

Zhong, R., Zhao, T., and Chen, X.: Hydrological model calibration for dammed basins using satellite altimetry information, Water Resour. Res., 56, e2020WR027442, doi: 10.1029/2020WR027442, 2020.

Zhang, S., Gao, H., and Naz, B. S.: Monitoring reservoir storage in South Asia from multisatellite remote sensing, Water Resour. Res., 50, 8927–8943, doi: 10.1002/2014WR015829, 2014.

Song, C., Huang, B., and Ke, L.: Modeling and analysis of lake water storage changes on the Tibetan Plateau using multi-mission satellite data, Remote Sens. Environ., 135, 25–35, doi: 10.1016/j.rse.2013.03.013, 2013.

Liu, J., Jiang, L., Zhang, X., Druce, D., Kittel, C. M. M., Tøttrup, C., and Bauer-Gottwein, P.: Impacts of water resources management on land water storage in the North China Plain: Insights from multi-mission earth observations, J. Hydrol., 603, 126933, doi: 10.1016/j.jhydrol.2021.126933, 2021.

Li, Yao, et al. "A high-resolution bathymetry dataset for global reservoirs using multi-source satellite imagery and altimetry." Remote Sensing of Environment 244 (2020): 111831.

While the hypsometric relationship from a DEM is practical and would also limit the need for water surface data, but not appropriate for our study. We give some reasons below.

- 1. Firstly, we agree with you that SRTM-DEM have problems in providing reliable bathymetry and hypsometric relationships for reservoirs built before 2000 (SRTM-DEM was developed in 2000, and most reservoirs are in high-fill state at that time) (Vu, D. T., HESS, 2022; Li et al., RSE, 2021). Many studies used SRTM-DEM to construct hypsometric relationships for reservoirs built after 2000 (Vu, D. T., HESS, 2022; Bonnema et al., WRR 2016). Although SRTM-DEM developed in 2000 can provide detailed hypsometric relationships for reservoirs built after 2000, in our study, only 161 out of 923 reservoirs recorded in GRanD database are built after 2000, and in our final retained datasets, only 47 out of 338 reservoirs are built after 2000 (See reservoir attributes.xlsx in our datasets). Constructing hypsometric relationships from DEMs at a national scale is not appropriate for our study and cannot satisfy our needs, since we aim to provide comprehensive reservoir WSE, SWA and RWSC at a national scale. Other DEMs such as MERIT-DEM and NASADEM are new and more likely to fail provide real detailed bathymetry of reservoirs.
- 2. Secondly, we adopted an enhancement algorithm developed by Zhao and Gao (2018) to map monthly SWA dynamics. The data can cover near all Chinese reservoirs and are evaluated as good performance in our study as well. Following our procedures described in Section 3.3, the constructed hypsometry relationships are mainly to

calculate RWSC, which are evaluated against in-situ datasets and verified as highquality datasets. In this sense, we could say that relationships derived in this study are reliable.

3. Thirdly, we aim to provide comprehensive reservoir datasets including water level, area and RWSC, thus, to make full use of satellite datasets (area, water level) for calculation can provide references in the background of booming satellites. We also cite the code and reference (Vu, D. T., 2022, hess; zenodo), and we will construct hypsometric relationships for these 47 reservoirs built after 2000 based on SRTM-DEM in our datasets as an alternative.

Hope above responses can address your questions.

**References:**

Vu, D. T., Dang, T. D., Galelli, S., and Hossain, F.: Satellite observations reveal 13 years of reservoir filling strategies, operating rules, and hydrological alterations in the Upper Mekong River basin, Hydrol. Earth Syst. Sci., 26, 2345–2364, https://doi.org/10.5194/hess-26-2345-2022, 2022.

Dung Trung Vu. (2022). Codes and Data of Satellite Observations Reveal 13 Years of Reservoir Filling Strategies, Operating Rules, and Hydrological Alterations in the Upper Mekong River Basin. https://doi.org/10.5281/zenodo.6299041

Bonnema, M., Sikder, S., Miao, Y., Chen, X., Hossain, F., Ara Pervin, I., Mahbubur Rahman, S. M., & Lee, H. (2016). Understanding satellite-based monthly-to-seasonal reservoir outflow estimation as a function of hydrologic controls. Water Resources Research, 52, 4095–4115. https://doi.org/10.1002/2015WR017830

Yamazaki, Dai, et al. "A high-accuracy map of global terrain elevations." Geophysical Research Letters 44.11 (2017): 5844-5853.

Li, Yao, et al. "Constructing Reservoir Area–Volume–Elevation Curve from TanDEM-X DEM Data." IEEE journal of selected topics in applied earth observations and remote sensing 14 (2021): 2249-2257.

2. It looks like many reservoirs have a negative value of storage (Figure 7). What further confuses me is that the gauged data have also negative values. How do you explain this matter (for both estimated and gauged data)? Shouldn't this problem be corrected? And wouldn't a more precise hypsometric relationship help?

R1C2: Thank you for the comment. We would like to clarify that we provide storage variations in our datasets, i.e., reservoir water storage change (RWSC), not the storage. Sorry for this misunderstanding and hope this can address your questions.

As already mentioned in the introduction, RWSC is an important variable that directly reflects the change of water stored in the reservoir, and can be implemented into global hydrological/hydrodynamic models for better streamflow simulation. Our plan is to fill a data gap, i.e., the remotely sensed reservoir water level, area, and RWSC in China, which can be applied as constraints to calibrate models or directly used for reservoir analysis. We listed all previous studies producing these three types of datasets in Table 1. Furthermore, in sections 4.2 and 4.3, we demonstrated the blueprint applications of our datasets, taking the Ankang reservoir as a case study, to show the value of our RWSC data in estimating reservoir release. RWSC can also be used to develop a reservoir storage forecast system at 1- to 3-month lead that can be valuable for water resource management in China.

We also agree with that storage is also important, but out of the scope of this study. Anyway, we are willing to give some points to this. Firstly, previous studies (and our study) mainly focused on developing RWSC rather than storage (Table 1). This can be attributed to the fact that the state-of-the-art of estimating accurate storage data need the accurate reservoir bathymetry, which is difficult to obtained from satellites. Secondly, although some studies make a good attempt, for example, Li et al. (2020) developed a bathymetry dataset for ~400 global reservoirs, the application is limit to large reservoirs which are observed by IceSat-2 mission, that has a coarse spatial resolution! So, it is impossible for us to adopt their methodologies to produce the remotely sensed storage for a large number of reservoirs in China.

**References:**

Li, Y., Gao, H., Zhao, G., and Tseng, K. H.: A high-resolution bathymetry dataset for global reservoirs using multisource satellite imagery and altimetry, Remote Sens. Environ., 244, 111831, doi: 10.1016/j.rse.2020.111831, 2020.

3. The quality of the presentation (including figures) could be enhanced. Please refer to my comments below.

R1C3: Thank you for the comment. The manuscript and figures are improved accordingly. Please find the responses below.

4. Are the water level and storage data retrieved from http://xxfb.mwr.cn/index.html available in the repository? Please correct me if I am wrong, but I couldn't find them. If that's true, I would encourage to authors to share those—it is not possible to download them from the aforementioned website.

R1C4: Yes, you are right. The in-situ datasets are updated day-by-day, thus, not possible to download the historical time series. I apologize for not making our collected in-situ datasets publicly available on Zenodo as we have a federal grant that limits the sharing of in-situ dataset. Moreover, we have no right to make all of them publicly available, now. Anyway, we are happy to share some data for users to do some case studies, please feel free to contact the corresponding author.

**General Comments:**

- Line 60 ("It is obvious that ..."). This sentence is not clear. Are you referring to China? If yes, I would state it clearly.

Changed as: Obviously, there is a data gap with regard to comprehensive reservoir information in China.

- Line 61-62. I suggest being more precise here. What are the reservoirs for which data are already available? Are the data public? And, importantly, what type of data are available?

Changed as: Records of a few Chinese reservoirs are available from these databases or previous studies (Table A1). Taking reservoir water level as an example, approximately 30 Chinese reservoirs are available from three datasets (Hydroweb, G-REALM and DAHITI).

| Data
type | No. of
reservoirs | Data sources             | Time and
temporal resolution    | Link                                                    |
|--------------|----------------------|--------------------------|------------------------------------|---------------------------------------------------------|
| Н            | 32                   | Hydroweb                 | 1992–2021, 10–35 day               | http://hydroweb.theia-land.fr/                          |
| Н            | 8                    | DAHITI                   | 2002–2021, 10–35 day               | https://dahiti.dgfi.tum.de/en/                          |
| Н            | ~30                  | G-REALM                  | 1992–2021, 10–35 day               | https://ipad.fas.usda.gov/cropexplorer/global_reservoir |
| Н            | <10                  | Tortini et al. (2020)    | 1992–2018, sub-
monthly/monthly | https://doi.org/10.5067/UCLRS-GREV2                     |
| А            | /                    | Bluedot                  | 2016–2021, sub-
monthly         | https://blue-dot-observatory.com/                       |
| А            | 923                  | GRASD                    | 1984-2018, monthly                 | https://doi.org/10.18738/T8/DF80WG                      |
| А            | ~8                   | Yao et al.
(2019)     | 1992–2018, sub-
monthly/monthly | https://lakewatch.users.earthengine.app/view/glats      |
| А            | 24                   | Liu et al.
(2020)     | 2004–2020, monthly                 | not publicly accessible                                 |
| А            | <10                  | Tortini et al.
(2020) | 1992–2018, sub-
monthly/monthly | https://doi.org/10.5067/UCLRS-AREV2                     |
| V            | <4                   | Busker et al.
(2019)  | 1984–2015, monthly                 | not publicly accessible                                 |
| V            | 24                   | Liu et al.
(2020)     | 2004–2020, monthly                 | not publicly accessible                                 |
| V            | <10                  | Tortini et al.
(2020) | 1992–2018, sub-
monthly/monthly | https://doi.org/10.5067/UCLRS-STOV2                     |
| V            | 10                   | Vu et al.
(2022)      | 2008-2020, monthly                 | https://doi.org/10.5281/zenodo.6299041                  |
| A, H, V      | 338                  | Shen et al. (2021)       | 2010-2020, monthly                 | https://zenodo.org/record/5812012                       |

| Table A1. Summary of recent studies and databases producing the remotely-sensed data on surface water area (A), water |
|-----------------------------------------------------------------------------------------------------------------------|
| surface elevation (H), and storage variation (V) in China.                                                            |

- Line 74. What do you mean with "difficult to be accessed"? Can they be accessed?

No, they are not openly accessible, but probably accessible upon request? We rephrase this sentence as: Moreover, the remotely-sensed datasets (e.g., lake/reservoir storage variations by Busker et al., 2019 or RWSC by Avisse et al., 2017) are not publicly available.

- Line 64-85. Vu et al. (2022) has just released a water level, surface, and storage dataset for 10 reservoirs in the Lancang Basin, China, for the period 2008-2020. This dataset was created using satellite data and modelling techniques similar to the ones reported here, so this is why I'm mentioning that study. Please note I'm a co-author of that paper, so please feel free to discard my comment.

Sorry for missing this new reference, and much thanks for your work and contributions. We added this in our Introduction, Table 1, Table A1.

**References:**

Vu, D. T., Dang, T. D., Galelli, S., and Hossain, F.: Satellite observations reveal 13 years of reservoir filling strategies, operating rules, and hydrological alterations in the Upper Mekong River basin, Hydrol. Earth Syst. Sci., 26, 2345–2364, https://doi.org/10.5194/hess-26-2345-2022, 2022.

- Table 1 is very informative (and I would leave it as is); however, it somewhat mixes studies and datasets that have different geographical foci and intents (e.g., global v. regional). I would therefore suggest including another table specifically focussed on China. It will help readers understand what is currently available—and how this study complements the state-of-the-art.

We created Table A1 specifically focussed on Chinese reservoirs. See above response.

- Line 111. "Testbed"?

Changed.

- Line 112-113. This sentence is not clear.

We rephrase this sentence as: Furthermore, to densify reservoir water level observations, merging data from multiple altimetric missions is meaningful given that satellite altimetry tracks are sparse and not available for all reservoirs (Jiang et al., 2019; Li et al., 2019).

**References:**

Jiang, L., Nielsen, K., Dinardo, S., Andersen, O. B., and Bauer-Gottwein, P.: Evaluation of Sentinel-3 SRAL SAR altimetry over Chinese rivers, Remote Sens. Environ., 237, 111546, doi: 10.1016/j.rse.2019.111546, 2020.

Li, Xingdong, et al. "High-temporal-resolution water level and storage change data sets for lakes on the Tibetan Plateau during 2000–2017 using multiple altimetric missions and Landsat-derived lake shoreline positions." Earth System Science Data 11.4 (2019): 1603-1627.

**- Section 2.1. How about the Repeat cycle of SARAL/AltiKa?**

We added this information Table 2. The SARAL/AltiKa satellite flew on the same repeat orbit as ENVISAT with a 35-day repeat cycle until July 2016, and was then switched to drifting orbit mode.

- Equations (1) and (2). Which technique did you use to estimate the various corrections? Were these corrections applied uniformly to all reservoirs or were they site-specific?

The different re-tracking algorithms mentioned in Table 1 are used to correct the Rrange in Equation (1). While the remaining corrections in Equation (2) such as geophysical and atmospheric corrections are directly taken from the official altimetry products. We added this information in our main text. Following the official user-guideline, they are site-specific.

- Line 178. I would say a few words about the algorithm developed by Zhao and Gao (2018). Also, is the code available?

We added a few words: This algorithm filled the gaps in area time series when the contamination in a Landsat image is between 5-95%, and applied interpolation and extrapolation for the missing monthly area estimates (i.e., no images or >95% invalid data). For more information and GEE code, please refer to Zhao and Gao (2018) and Shen et al. (2022).

The code is not available from Zhao and Gao, but we can make our written code available. Please contact the first author, Youjiang Shen (yjshen2020@gmail.com)! Once the paper accepted, the code will appear on Zenodo as well (the same link).

- Figure 1. I suggest improving / re-drawing Figure 1. It's very hard to visualize the

reservoirs (pink squares). Also, the colour-bar for the elevation is missing.

See the modified Figure 1 below.

---

## Author Response (AR1)

Dear Handling topical editor David Carlson,

Thank you so much for your work in handling with our manuscript "essd-2021-470". We are very happy to hear the positive feedback on our datasets from community members (see the https://doi.org/10.5281/zenodo.7251283) and referees. We have carefully studied all comments and made corrections/changes as suggested.

Here, we briefly summarize the improvements to our dataset as well as the main changes in our revised manuscript considering the minor and moderate issues raised by the reviewers.

- Regarding the presentation of this manuscript, we have rewritten almost the entire manuscript and have tried to help the readers to better understand our study.
- We performed strict data quality control to eliminate any potentially problematic data points, and data version 2 is released as the final dataset for our manuscript that is under consideration for being published in ESSD.
- For water levels, we added six satellite altimeters (previsous version with four satellite altimeters) in this version and provided two modes, standardmeasurement, and enhanced-measurement. For the enhanced-measurement product, we need to eliminate the systematic bias between satellites. The biases are removed using two methods, as detailed in comment R2CO, to address the main issue raised by Reviewer 2.
- For water area, we adopted a new monthly surface water area extent (SWA) from Landsat and Sentinel-2 images by using a new algorithm developed by Donchyts et al. (2022) and added a section to describe the accuracy assessment to evaluate our area dataset based on in situ water levels in 93 reservoirs, our generated altimetric water levels and cross validation against the other two existing area datasets to address the main issue raised by reviewers 2 and 3.
- For water storage variation, we added DEM-based estimates, so that we provide two types of storage variations, i.e., using satellite-based water levels and water area, and using satellite-based water area and DEM-based curves. The storage changes are validated against in situ storage changes for 93 reservoirs, addressing the main issue raised by reviewer 1 and 3 (Please note the major issue raised by reviewer 1 why we have negative values for storage variation dataset, because he incorrectly assumes that we provide storage data.)

Here, we would like to clarify the novelty of our data product again. In our revised manuscript, we generate (1) reservoir water level (WSE) from six satellite altimeters (Sentinel-3 A/B, CryoSat-2, SARAL/Altika, Jason-3, and ICESat-2), and provide two modes of WSE, standard measurement (single satellite) and enhanced measurement

(i.e., merging measurements from multiple satellites for a specific reservoir if possible); (2) monthly surface water area extent (SWA) from Landsat and Sentinel-2 images by using a new algorithm developed by Donchyts et al. (2022); (3) monthly reservoir water storage change (RWSC) data and provide two types of RWSC (i.e., one type using satellite-based water levels and water area, and another type using satellitebased water area and DEM-based area-storage curves). Please note the following four points highlighting the novelty and value of our datasets: (a) in our revised data and manuscript, we performed strict data quality control to eliminate any potentially problematic data points, and performed a validation using 93 reservoirs with in situ water level and storage data, which is not done ever before for Chinese reservoirs; (b) although there are some similar databases covering partly reservoirs in our study, however, RWSC developed by Hou et al. (2022) are not publicly available, the methodology is also different, Hou et al. (2022) relied the GRSAD water area data and DEM (a geo-statistical approach) to calculate the storage change. Our datasets are publicly available and we provide two types of RWSC and validated their performance, which could be very valuable for relevant studies; through the literature review and our statistical metrics, the RWSC estimates from satellite WSE-SWA are more reliable than those from DEM-Area approach (c) we noted that there are several databases providing SWA such as GRSAD (Zhao and Gao, 2018), RealSAT (Khandelwal et al., 2022) and Donchyts et al. (2022), and WSE data; however, whether SWA/WSE time series from different databases have a good agreement with one another and gauged measurements is not systematically evaluated, which can be shown in this study; (d) moreover, a growing interest in using satellite altimetry data in hydrological cycle is expected, thus knowing the accuracy of satellite altimetry is a prerequisite, although previous studies assessed satellite altimeters in retrieving reservoir water levels (Shu et al., 2021), knowledge is still limited as to evaluations of different altimeters for a large sample of reservoirs, which can be shown in this study.

Hope the revised manuscript and data are to your satisfaction and meet the standard of ESSD journal.

Best,

Youjiang Shen.

**Reviewer #1 Comment on essd-2021-470 (Stefano Galelli)**

**Dear Stefano Galelli,**

Thank you for your time and efforts in reviewing our manuscript. We are very happy to hear your positive feedback on our datasets which provide strong support for many aspects. We are very happy about the agreement that the presented data set is an important contribution to the large-scale studies that will take place in the context of increasing activity in the reservoir and dam sector and are pleased to be able to contribute with this data set. Please find attached point-to-point responses regarding your comments (marked in purple) and made corresponding changes in the main manuscript (in red). We hope that the improved manuscript can help the readers to better understand our study.

**Kind regards.**

**General comments**

Manuscript essd-2021-470 describes a novel dataset providing water surface, level, and storage information for 338 reservoirs in China. In my opinion, this is a muchneeded dataset that fills in an important gap, since data on water reservoirs are typically not available to the international community. I believe many studies and downstream applications will thus benefit from these data.

**R1CO: Thank you for your positive comments identifying the strengths of our work.**

Overall, both manuscript and dataset are well organized, although a few important probably deserve more attention. In particular:

1. I am not entirely convinced about the approach used to estimate the hypsometric relationships, which, if I understand correctly, are based on water level and surface data estimated from satellite data. In general, water level data are rather reliable, while it is always a challenge to get the right water surface data (a matter that explains the use of image enhancing techniques), a problem that might affect the quality of the curves. So, why not using a DEM to get the right curves? This could be done for many reservoirs. Estimating the hypsometric relationships from a DEM would also limit the need for water surface data.

R1C1: We added this DEM approach in the revised manuscript, thus, we provided two types of **RWSC estimates**, i.e., one is to use water level and water areas from satellite altimeters and images, while another one is to use imagery-based water areas and DEM (digital elevation model). The core of these two approaches is to construct the A–E relationships from the overlapping records of water level and areas or DEM. The

performance of these two types of RWSC estimates are shown in Section 3.3. Here, we give the corresponding changes in the revised manuscript.

**2.3 Reservoir storage variation estimation**

[revised manuscript text omitted]

---

## Author Response (AR2)

Dear Handling topical editor David Carlson,

We are very happy to hear the positive feedback on our datasets and thank you so much for your great work and insightful comments for improving our manuscript. Please find the attached response to the questions raised by your side and small suggestions by third referee. We hope the revised texts and responses are to your satisfaction.

Best,

Youjiang Shen.

**Review by editor (David Carlson)**

**Comments to the author**:
Thank you for sharing products of complex data processing effort.

I read some residual language uncertainties, due perhaps to Chinese-English translation. I assume Copernicus language experts will solve these. Authors will need to provide close attention and frequent correspondence during proofing steps.

E1C0: Thank you so much for improving our manuscript and we will pay close attention on these language uncertainties and give frequent correspondence during proofing steps.

Three questions remain as we move forward:

1) Authors frequently make the point that these data cover 50% of national water storage capacity. What about remaining 50%. Why these? What about other 'missing' data. Do authors plan future studies on other 50%?

E1C1: Thank you for your question. This issue is mainly due to the low spatial coverage of current satellite datasets. We need to be aware that the satellite altimetry ground track is rather sparse. The majority of the remaining 50% reservoirs are not crossed by the ground tracks. Therefore, no altimetry data is available. Future satellite missions such as SWOT mission with higher resolution may fill the data gap. Yes, if future products cover these reservoirs, we will update this dataset.

The corresponding revised texts are marked in red in our texts (Section 5) and copied here for your convence.

Nonetheless, for reservoirs accounting the remaining 50% water storage capacity, current satellite altimetry missions are not able deliver enough useful observations or detect these reservoirs given by their sparse altimetric ground tracks, therefore, not included in our products. Developing more general algorithms with better performance regardless of the reservoir's attributes and using satellite altimetric data with higher temporal resolution (e.g., Surface Water and Ocean Topography mission, Biancamaria et al., 2016) will be our next studies. Overall, our study fills such a data gap by incorporating various satellites into a comprehensive reservoir data set at national

scale. We envision this dataset can be immediately applied to some scientific areas described in Sect. 4, and can provides strong support for many aspects such as hydrological processes and water management studies.

References:

Biancamaria, S., Lettenmaier, D. P., and Pavelsky, T. M.: The SWOT mission and its capabilities for land hydrology, Surv. Geophys., 37, 307–337, doi: 10.1007/s10712-015-9346-y, 2016.

2) Authors say most reservoirs lie in eastern and central China. Map in Figure 1 shows distribution but also raises a classification issue. With good latitudinal coverage but relatively narrow longitudinal coverage, most reservoirs lie in lower (elevation and flow) regions, with hydrological impacts? Understand about availability of in situ gauge data but, related to question about other 50% above, what biases if any do authors expect from somewhat restricted geographic focus. E.g. if most of this water used for irrigation rather than power generation, then users should expect different seasonal patterns of water withdrawal. Withdrawal patterns might vary upstream vs downstream. As a caution to users, authors need to more explicitly defend their selections?

E1C2: Thank you for your question. Hope the above response partly addressed your questions. Firstly, this is true that Chinese reservoirs are unevenly distributed across the nation, i.e., most reservoirs lie in eastern and central China. And our products show the similar distributing pattern. Regarding our selections of these reservoirs, still, this issue is mainly due to the availability of satellite datasets or in situ gages. Therefore, only reservoirs covered by satellite altimetry datasets are selected for further processing. For the main use of reservoirs in our products, their attributes are summarized in datasets reservoir_attributes.xlsx.

3) Authors allow confusion about 338 vs 93 reservoirs. Help readers better understand validation processes on 93 and extension of those to 338?

E1C3: Again, it is mainly due to the availability of in situ data. We try to help readers to better understand this validation processes. In the section 2, we firstly introduce the initial reservoirs that are used for processing and our collected in situ datasets.

In this study, we selected all reservoirs for which geographical information is available from the GRanD database (http://globaldamwatch.org/grand/, Lehner et al., 2011). We obtained daily water level and storage data spanning 2015–May 2021 for 93 reservoirs from the local watershed agency (http://xxfb.mwr.cn/index.html) and National Hydrological Information Centre for validation (http://113.57.190.228:8001/web/Report/BigMSKReport).

Followed by the methodologies, we then give a data description in section 3.1.

In this study, we generated the remotely sensed reservoir datasets for 338 Chinese reservoirs, with a total of 470.6 km$^3$ storage capacity (50% reservoir water capacity in China). In the directory of 01_res_loc, we provide two ESRI shapefiles (the location of

338 reservoirs and 93 reservoirs with in-situ observations for validation) and one Excel file describing their associated attributes.

In section 3.2, we explained why 338 reservoirs are finally retained in our products.

In total, 921 reservoirs are visited by the six altimetry missions over China during CryoSat-2 era, providing basic WSE information. After outlier removal, time series construction and combination, and visual inspection, we finally retain 338 reservoirs that have enough valid measurements. Note that, most reservoirs are removed due to the insufficient altimetry data points rather than other reasons.

4) Zenodo data link shows two versions of 01 res_loc files. One apparently for MACOSX? Good description of upper files but please also explain (or remove) lower files?

E1C4: Thank you for your suggestions. I deleted the MACOSX version and checked the data set. A more detailed description of is enclosed in ReadMe file, and my email/(future published DOI) is also posted on Zenodo so that users can contact me if any inconvenience to download these datasets happens.

**Review by third referee (Anonymous Referee)**
I appreciate the authors' great efforts in addressing my comments. The manuscript has been much improved compared to the previous version. I think it is almost ready for publication in ESSD. Before that, I would suggest clarifying what kind of A-E relationships (e.g., R2 threshold) have been considered as valid relations to estimate reservoir storage variations in your product. Because storage estimates would be useless if they were based on poor A-E relationships.

R3C0: Thank you for your insightful comments for improving our manuscript. This question has been also explained in response 5 to second referee.

Firstly, we will say that 0.5 $R^2$ threshold value is used as valid relations to estimate reservoir storage variations in our product, which is also used in previous studies (Busker et al., 2019; Li et al., 2020).

Secondly, in our revised dataset, we performed strict data quality control to eliminate any potentially problematic data points, resulting a reliable A–E curve for most reservoirs (84%) with $R^2 > 0.5$.

Thirdly, we used in-situ observations of 91 reservoirs as an important reference to validate RWSC dataset, thus bringing the good level of confidence in our data quality. Furthermore, we provided monthly RWSC time series from 2010 to 2021 in two modes: one is to use WSE and SWA from satellite altimeters and images, while another one is to use satellite SWA and area-storage model developed by DEM, as suggested by referee 1.

References:

Busker, T., de Roo, A., Gelati, E., Schwatke, C., Adamovic, M., Bisselink, B., Pekel, J.-F., and Cottam, A.: A global lake and reservoir volume analysis using a surface water dataset and satellite altimetry, Hydrol. Earth Syst. Sci., 23, 669–690, doi: 10.5194/hess-23-669-2019, 2019.

Li, Y., Gao, H., Zhao, G., and Tseng, K. H.: A high-resolution bathymetry dataset for global reservoirs using multi-source satellite imagery and altimetry, Remote Sens. Environ., 244, 111831, doi: 10.1016/j.rse.2020.111831, 2020.